# Loss-to-Loss Prediction: Scaling Laws for All Datasets

**David Brandfonbrener**                              *david.brandfonbrener@gmail.com*
*Kempner Institute, Harvard University*

**Nikhil Anand**                                     *nikhil_anand@harvard.edu*
*Kempner Institute, Harvard University*

**Nikhil Vyas**                                      *nikhil@g.harvard.edu*
*SEAS, Harvard University*

**Eran Malach**                                      *eran.malach@gmail.com*
*Kempner Institute, Harvard University*

**Sham Kakade**                                      *sham@seas.harvard.edu*
*Kempner Institute and SEAS, Harvard University*

**Reviewed on OpenReview:** *https://openreview.net/forum?id=1Avb4jYjLb*

## Abstract

While scaling laws provide a reliable methodology for predicting train loss across compute scales for a single data distribution, less is known about how these predictions should change as we change the distribution. In this paper, we derive a strategy for predicting one loss from another and apply it to predict across different pre-training datasets and from pre-training data to downstream task data. Our predictions extrapolate well even at 20x the largest FLOP budget used to fit the curves. More precisely, we find that there are simple shifted power law relationships between (1) the train losses of two models trained on two separate datasets when the models are paired by training compute (train-to-train), (2) the train loss and the test loss on any downstream distribution for a single model (train-to-test), and (3) the test losses of two models trained on two separate train datasets (test-to-test). The results hold up for pre-training datasets that differ substantially (some are entirely code and others have no code at all) and across a variety of downstream tasks. Finally, we find that in some settings these shifted power law relationships can yield more accurate predictions than extrapolating single-dataset scaling laws. [1]

## 1 Introduction

Scaling laws ([Kaplan et al., 2020](); [Hoffmann et al., 2022]()) have become a reliable tool for extrapolating model performance (as measured through, e.g., cross-entropy loss on held-out data), as well as a way to determine optimal model size given a FLOP budget ([Llama 3 Team, 2024]()). In their standard form, scaling laws essentially predict the training loss for a given model size and dataset size. However, these scaling laws are distribution-dependent and only apply to the training distribution that is used to fit the scaling law. Relatively little is known about how they change across different pre-training distributions, and how to use scaling laws to predict transfer performance on downstream test distributions.

In this paper, we take a first step towards understanding how scaling laws change as we change either the training distribution or the testing distribution. To do this, we propose loss-to-loss prediction, a methodology for predicting the loss on one data distribution from the loss on another. This is useful since once we have a function that predicts one loss from another, we can take a scaling law fit on the first loss and immediately

---

[1] Notebooks: https://github.com/KempnerInstitute/loss-to-loss-notebooks,
Training code: https://github.com/KempnerInstitute/loss-to-loss-olmo,
Models: https://huggingface.co/KempnerInstituteAI/loss-to-loss

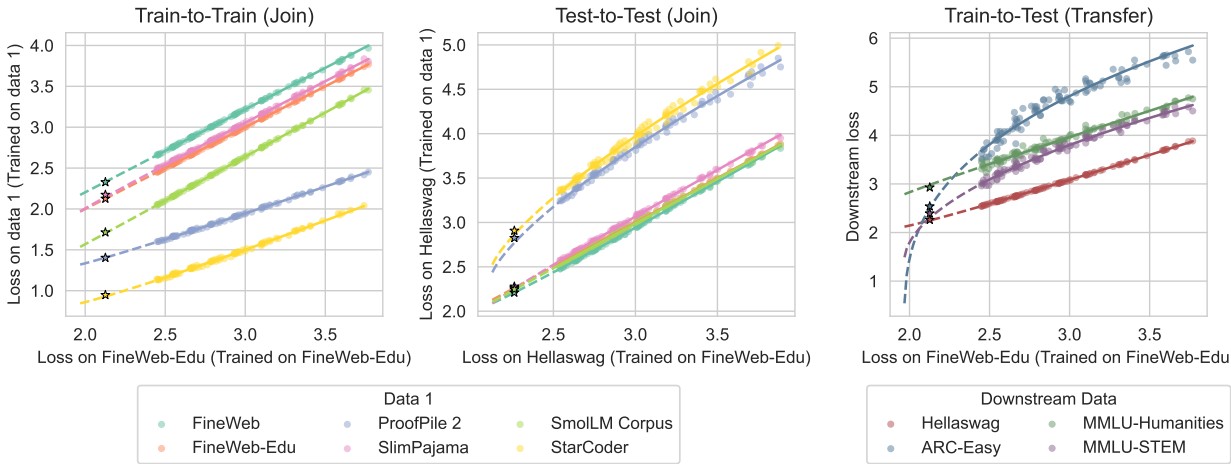

Figure 1: (Left) Train-to-train prediction from FineWeb-edu to all 6 training sets. Each datapoint represents a pair of models that are "joined" on model size $N$ and dataset size $D$. Dashed lines represent extrapolation and stars represent 3.3B models trained with 20x compute of the largest dot. These large models are *not* used to fit the curves. (Center) Test-to-test prediction of Hellaswag cross entropy loss between models trained on FineWeb-edu and models trained on the other datasets. Again each datapoint represents two models joined on model and dataset size. The downstream loss is the cross entropy loss of the correct answer to the multiple choice problem when phrased as a cloze task. (Right) Train-to-test prediction from FineWeb-edu to four downstream tasks. Each datapoint represents a single model and its "transfer" performance on the val data.

translate it to a scaling law for the second loss. Further, if we have a suite of models trained on one dataset and want to predict the performance we would get from training on a new dataset, we can apply loss-to-loss prediction. Moreover, there is an independent scientific question of how scaling laws change across datasets.

Our main results are the observations of three types of loss-to-loss relationships shown in Figure 1. First, we consider train-to-train, comparing training loss across models trained on two different datasets. When models are paired by training compute we find that there is a shifted power law that relates the two losses. This has implications for how scaling laws vary across datasets and for being able to predict new scaling laws from smaller samples by translating existing scaling laws from other datasets.

Second, we consider train-to-test transfer where a model trained on one dataset is evaluated on a different dataset. Again, we find that a shifted power law is predictive (although with a slightly different shift). These results are less useful for prediction, since they do not predict performance on new train sets. However, they have implications for understanding how pre-training transfers to downstream tasks.

Third, we consider test-to-test prediction where we compare downstream test loss across models trained on two different datasets. Like train-to-train prediction, we find a shifted power law when pairing models by model and dataset size. These results are noisier than the others, but have implications for selecting data to improve performance on downstream tasks.

Finally, we consider applications of these relationships to a practical setting. In this setting, a scaling law has already been fit on one dataset and we wish to make some prediction about what will happen on a new dataset given only a very small number of training runs on the new dataset. Explicitly, we look at two types of prediction in this setting. First, we consider a setting where we want to fit a scaling law on a new training set and show that leveraging train-to-train predictions can yield substantially better predictions with as few as eight models trained on the new dataset. Second, we consider predicting the test performance of a larger model trained on the new dataset and find that test-to-test prediction can yield better predictions than extrapolating from runs on the new dataset alone.

To summarize, our main contributions are:

- We derive a methodology for loss-to-loss prediction that translates scaling laws between datasets.

- We illustrate train-to-train, train-to-test, and test-to-test prediction across pre-training datasets on 6 diverse pre-training datasets and 11 downstream tasks. We discuss implications for understanding scaling laws, transfer learning, and generalization to downstream tasks.

- We show that leveraging data from multiple pre-training datasets can yield better predictions about what will happen when training on new datasets than fitting independent scaling laws.

## 2 Related work

### 2.1 Scaling laws

Standard approaches to scaling laws attempt to fit a curve to the optimal number of model parameters $N$ and training tokens $D$ to minimize the *pre-training loss* under a given budget of FLOPs (Hestness et al., 2017; Kaplan et al., 2020; Hoffmann et al., 2022; Porian et al., 2024; Abnar et al., 2021; Maloney et al., 2022; Bordelon et al., 2024a).

To fit these curves, it is useful to specify a parametric form of the loss in terms of $N$ and $D$. Hoffmann et al. (2022) assumes this curve takes the following form:

$$L(N, D) = E + \frac{A}{N^\alpha} + \frac{B}{D^\beta}. \tag{1}$$

This formula is inspired by classical upper bounds on a loss decomposition that attributes error to Bayes risk (entropy), approximation error (from having finite parameters), and estimation error (from having finite data) (Bottou and Bousquet, 2007).

On the other hand Kaplan et al. (2020) instead assumes that:

$$L(N, D) = \left( \left( \frac{A}{N} \right)^{\alpha/\beta} + \frac{B}{D} \right)^\beta. \tag{2}$$

Below, we will advocate for this formulation, but with an added irreducible entropy term $E$:

$$L(N, D) = E + \left( \left( \frac{A}{N} \right)^{\alpha/\beta} + \frac{B}{D} \right)^\beta \tag{3}$$

Regardless of the functional form, scaling laws have been an integral part of the success of modern neural language models. Our work builds on the ideas originated in this line of work and extends them to consider how to translate scaling laws across data distributions.

### 2.2 Scaling laws for transfer and downstream tasks

Scaling laws for pre-training loss are useful as a proxy to guide pre-training, but we ultimately care about downstream task performance. Prior work attempting to tackle this issue has found that directly computing hard metrics like accuracy can lead to the appearance of emergent behaviors and suggests using softer metrics like cross entropy loss instead (Schaeffer et al., 2024a;b). This is corroborated by Du et al. (2024) which notes that while downstream accuracy can vary smoothly with training loss at some points in the curve, the hardness of the accuracy metric means that no progress in accuracy above random chance will be observed until some "emergent" loss level.

On the other hand, Gadre et al. (2024) claims that downstream accuracy can be predicted as a function of training loss with a similar exponential curve to the one we propose for predicting downstream loss. However, they only claim this is predictable when averaging over many tasks and carefully selecting which tasks to use. In this paper when considering downstream tasks we focus on single downstream tasks and find loss to be a more stable metric than accuracy. A detailed discussion of loss versus accuracy is in Appendix A.

Another related line of work comes from the distributional robustness literature on "accuracy on the line" (Miller et al., 2021; Tripuraneni et al., 2021; Awadalla et al., 2022). This phenomena focuses on the relationship

between the accuracy of a single model across two closely related tasks, like different versions of imagenet, and finds that accuracy on one will predict accuracy on the other. We consider loss rather than accuracy, language modeling rather than vision, and find non-linear fits.

Another line of work uses linear data models to predict performance for fixed model sizes when training on subsets of some larger set of data (Ilyas et al., 2022; Li et al., 2023a; Engstrom et al., 2024). In contrast, our work focuses on predicting losses across scales and across distinct distributions rather than subsets. Moreover, we find that we need simple non-linear models in our setting.

Note, in this work we focus on zero shot transfer where there is no finetuning on the target task. Prior work on "transfer scaling laws" focuses instead on a finetuning setting (Hernandez et al., 2021; Abnar et al., 2021; Isik et al., 2024), which is interesting, but beyond the scope of this work.

## 3 Setting

### 3.1 Notation

We are interested in studying transfer across different training distributions. To formalize this, we will define two distributions: $P_0$ and $P_1$. We will consider $P_0$ as the "source" and $P_1$ as the target. The goal is to use a function of the loss on $P_0$ to predict the loss on $P_1$. As an example, $P_0$ could be FineWeb and $P_1$ could be Starcoder or Hellaswag. We use $L_i$ to indicate the loss calculated on distribution $P_i$ (averaged per-token). If $P_1$ represents a multiple choice task, we will let $L_1$ be the loss of correct answer when the question is phrased as a cloze task (following (Schaeffer et al., 2024b; Madaan et al., 2024)).

Given a pre-training distribution $P_i$, we let $\hat{f}_i^{N,D}$ denote an $N$ parameter model trained on $D$ tokens sampled from $P_i$. Our results present comparisons across losses $L_0, L_1$ for models $\hat{f}_0^{N,D}, \hat{f}_1^{N,D}$ when sweeping across different choices of $P_0, P_1$, as well as $N, D$.

When we refer to a scaling law fit from Equation (3) on distribution $P_i$, we will append a subscript to the corresponding parameters. For example, the irreducible entropy of the scaling law fit on $P_0$ is denoted by $E_0$.

### 3.2 Experimental methodology

To facilitate our analysis, we pre-train models of varying size with varying flop budgets on 6 pre-training datasets: FineWeb (Penedo et al., 2024), FineWeb-edu (Penedo et al., 2024), Proof Pile 2 (Azerbayev et al., 2023; Computer, 2023; Paster et al., 2023), SlimPajama (Soboleva et al., 2023), SmolLM Corpus (Ben Allal et al., 2024), and Starcoder v1 (Li et al., 2023b). We train all models using OLMo (Groeneveld et al., 2024) and generally follow hyperparameter settings from Wortsman et al. (2023); Zhao et al. (2024). Importantly, we use a linear warmup and cosine decay schedule for every run and only report the final performance (Porian et al., 2024). For downstream evaluations, we directly evaluate the models zero-shot on the downstream tasks as cloze tasks (Schaeffer et al., 2024a). Full hyperparameters and details can be found in Appendix D.

FLOP budgets for our sweep range from 2e17 to 4.84e19 and model sizes range from 20M to 1.7B. The optimal model at the largest FLOP budget is roughly 750M (it varies per dataset). The total grid contains 528 models, or 88 models per training dataset. For our extrapolation experiments, we train 6 larger models (one for each dataset) at a FLOP budget of 1e21 each of size 3.3B. Full scaling law fits and illustrations of the full grid of model sizes are in Appendix C.

## 4 The hypothesis of loss-to-loss prediction

The main idea we propose in this paper is to use one loss to predict another. In this setup, each loss can be either train loss or test loss and the two losses can be for models trained on different datasets. Formally, we want to predict $L_i(\hat{f}_j^{N,D})$, the loss on distribution $P_i$ for a model trained on $P_j$, when given $L_k(\hat{f}_\ell^{N,D})$ where $k, \ell$ can be different from $i, j$. Essentially, we want to parameterize a function $g$ such that

$$L_i(\hat{f}_j^{N,D}) \approx g\left(L_k(\hat{f}_\ell^{N,D})\right). \tag{4}$$

**Hypothesis.** Our main hypothesis is to posit the following functional form for $g$:

$$L_i(\hat{f}_j^{N,D}) \approx K \cdot \left( L_k(\hat{f}_\ell^{N,D}) - E_{k|\ell} \right)^\kappa + E_{i|j}. \tag{5}$$

where $K, \kappa$ are free parameters and $E_{i|j}$ is the irreducible error in the scaling law of the loss $L_i$ (computed on the distribution $P_i$) for models trained on data from the distribution $P_j$.

**Motivation.** The functional form is inspired by a simple observation: both $L_i(\hat{f}_j^{N,D})$ and $L_k(\hat{f}_\ell^{N,D})$ obey scaling laws in $N$ and $D$. So $g$ needs to be a valid mapping between scaling laws, meaning that if we pass in a loss that obeys a scaling law as input to $g$ we get out a valid scaling law as output.

To see how our functional form respects this observation, we can start from two independent scaling laws:

$$L_i(\hat{f}_j^{N,D}) = E_1 + \left( \left( \frac{A_1}{N} \right)^{\alpha_1/\beta_1} + \frac{B_1}{D} \right)^{\beta_1}, \qquad L_k(\hat{f}_\ell^{N,D}) = E_2 + \left( \left( \frac{A_2}{N} \right)^{\alpha_2/\beta_2} + \frac{B_2}{D} \right)^{\beta_2} \tag{6}$$

Then, we can see that Equation (5) is a valid translation. In particular, we get the following relationships between parameters of the scaling laws under our parameterization of $g$:

$$\alpha_1 = \kappa\alpha_2, \quad \beta_1 = \kappa\beta_2, \quad A_1 = K^{\frac{1}{\kappa\alpha_2}} A_2, \quad B_2 = K^{\frac{1}{\kappa\beta_2}} B_2. \tag{7}$$

Note that there is not a similar way to transform the (Hoffmann et al., 2022) scaling laws of Equation (1) since we cannot modify the exponents without introducing cross terms that depend on both $N$ and $D$. We should caveat that while the (Hoffmann et al., 2022) formulation does not allow for valid translations, it does empirically give good and very similar fits to the (Kaplan et al., 2020) version. We think it is an interesting open question to precisely pin down the correct formulation for scaling laws and view our work as one data point supporting (Kaplan et al., 2020) with an added irreducible entropy term.

**Caveat.** We should caveat that while the hypothesis is stated for any four distributions $(P_i, P_j, P_k, P_\ell)$, we do believe that there could be adversarial distributions constructed so that the relationship no longer holds. In this work we focus on natural distributions of text and we find that the relationship does hold across surprisingly diverse data (e.g. from data that is entirely code to data that does not contain code at all), though there are likely limits to how far these relationships hold.

**Comment on theory.** In appendix G, we examine how existing theoretical models of scaling laws connect to our empirical findings. Using a simplified linear model with power-law feature spectra, we show that the model exhibits scaling behavior qualitatively similar to our proposed form. While this theoretical framework successfully captures some aspects of in-domain transfer, we note that it does not yet explain the full richness of empirical scaling behaviors, particularly for our out-of-distribution transfer results. This analysis suggests that while existing linear models provide useful insights, additional theoretical development may be needed to fully explain the observed phenomena in real datasets.

**Summary.** So far, this is a very general hypothesis. It says that given any pair of losses (which can be train losses or test losses) trained on any pair of data distributions we can fit two free parameters $K, \kappa$ to translate between the losses (given estimates of the irreducible entropy of each loss). In the following sections we make this more concrete and show how it can be used for train-to-train, train-to-test, and test-to-test prediction and how these predictions can be practically useful.

## 5 Empirical evidence for loss-to-loss prediction

In this section, we present the loss-to-loss relationships that for the core observation of the paper. In turn we will present train-to-train, train-to-test, and test-to-test relationships.

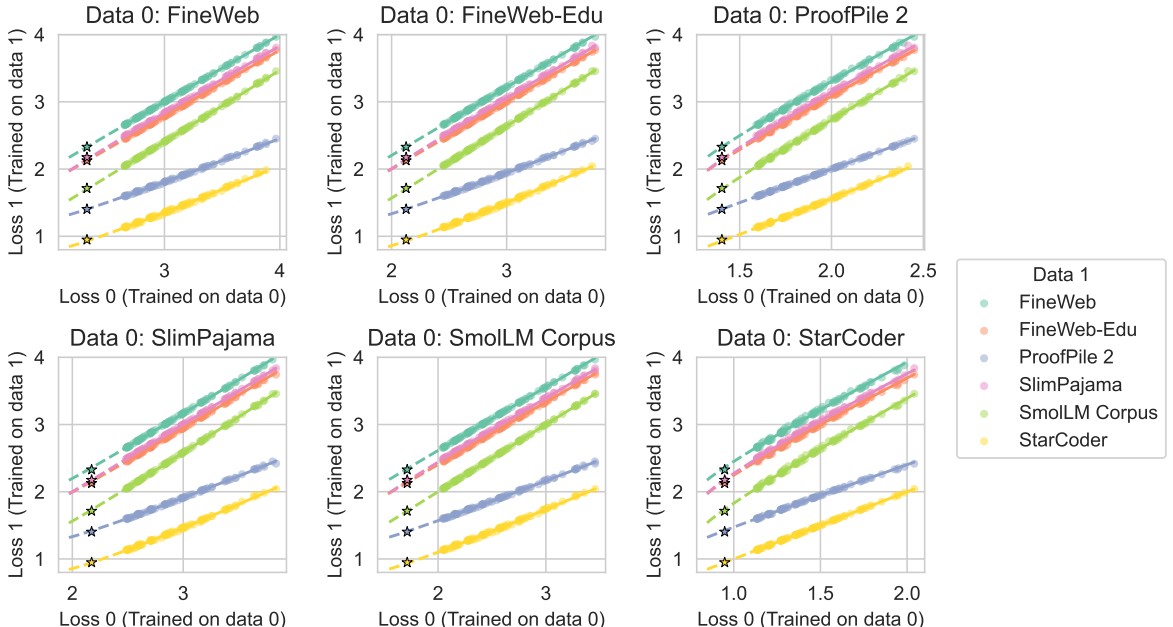

Figure 2: Train-to-train fits. Each point on the plot represents the final loss of two models: $\hat{f}_0^{N,D}$ which is trained on dataset 0 and $\hat{f}_1^{N,D}$ which is trained on dataset 1. The models are paired when they use the same number of parameters $N$ and tokens $D$. Starred points indicate a large model trained for the purpose of testing the extrapolation of the curves, which are only fit on the dotted points.

### 5.1 Train-to-train prediction

Our first main result is to observe a consistent scaling relationship between train losses across datasets. Explicitly, we find that by fitting just two parameters $K$ and $\kappa$ we can capture and extrapolate the scaling relationship between pairs of training losses as follows:

$$L_1(\hat{f}_1^{N,D}) \approx K \cdot \left(L_0(\hat{f}_0^{N,D}) - E_0\right)^{\kappa} + E_1 \tag{8}$$

Note, this is comparing *different* losses and *different* models, but the models are paired when they each have $N$ parameters trained on $D$ tokens. Also, recall that $E_0, E_1$ are the irreducible errors from *independent* scaling law fits on $P_0$ and $P_1$ respectively. Finally, note that since we are only fitting a slope and exponent, each curve is linear on a shifted log-log scale. However, since we are plotting 6 curves in one plot, each with different $E_1$, we cannot display them all consistently log-log plot and opt for a linear scale for clarity. Further details on the curve-fitting process are in Appendix F. Results for fitting these curves can be seen in Figure 2. We also include comparisons to linear fits rather than power laws in Appendix B.1.

**Compute optimal models.** Under the parameterization in Equation (8) for translating between losses, the size of the compute optimal is invariant. To see this, note that the optimal model size for a given flop budget $N^*(C)$ can be expressed as $(\frac{GC}{6})^a$ for $a = \frac{\beta}{\alpha+\beta}$ and $G = \frac{\alpha A^{\alpha/\beta}}{\beta B}$ under the assumption that $C = 6ND$. Coupled with the relationships described in Equation (7), this implies that under the transformations induced by Equation (8) the function $N^*(C)$ is invariant.

This implies that for a given FLOP budget, the optimal model size is the same for any data distribution where this translation relationship holds. This seems like a strong conclusion, but does fit in with common empirical practice after Hoffmann et al. (2022) where practitioners often train on approximately 20x more tokens than parameters in a model across datasets. Of course, if anything changes in the model architecture or training algorithm, then this translation and this invariance would not hold anymore, but under Equation (8) the compute optimal model size is invariant to changes in the data distribution. It is an interesting open

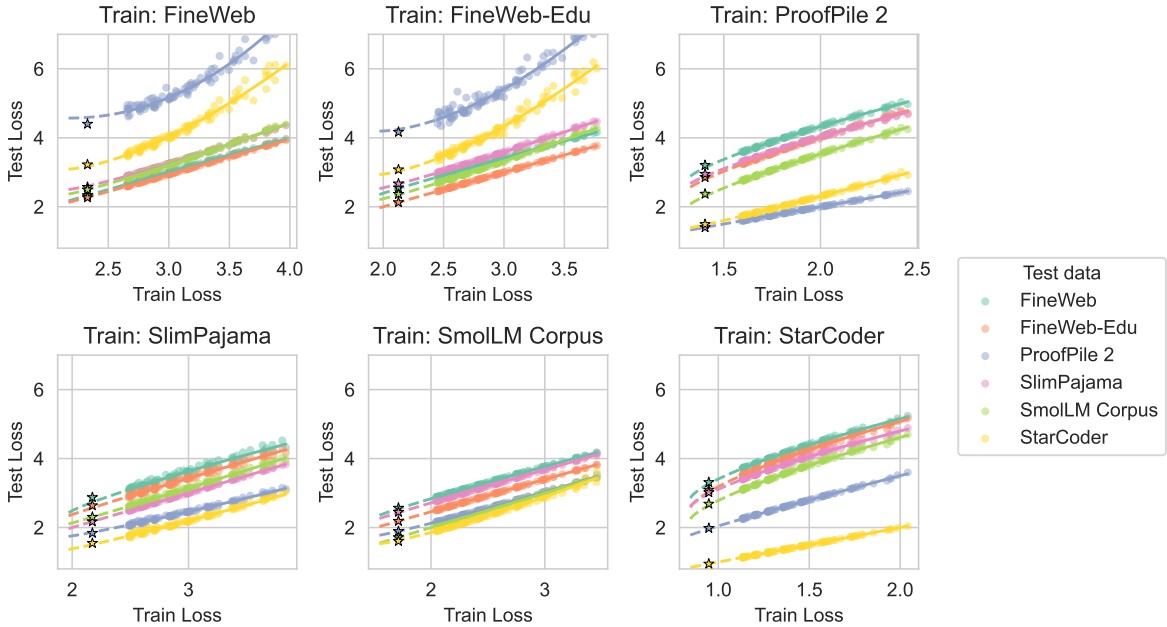

Figure 3: Train-to-test fits. Each datapoint represents a single model trained on the dataset in the subplot title and then evaluated on a different dataset as indicated by the color.

question to test how generally this invariance holds. It roughly holds across the 6 datasets we test which differ substantially (some are all code, others all English), but it may break down for some other dataset pairs.

**Parameter values.** Note that the exponents $\kappa$ tend to be close to 1. If $\kappa = 1$ for a pair of datasets, this means that they have the same scaling exponents. Across all pairs of datasets the minimum is 0.88 and maximum is 1.13, which occur between Starcoder and SlimPajama depending on the direction of prediction. While these are close to 1, these are sufficiently far enough from 1 that trying to make a linear fit will lead to substantially worse extrapolation predictions.

On the other hand, $K$ tends to be further from 1. There the largest differences come between SmolLM Corpus and ProofPile (either 0.55 or 1.72 depending on the direction of prediction). This suggests that the differences in returns to scale between datasets are clearly seen in differences in the numerators of the scaling laws. Further, note that it is interesting and not obvious a priori that we can fit just a single multiplicative constant $K$ which modifies both $A$ and $B$ in Equation (3).

**Implications.** In summary, the train-to-train prediction results have a few implications:

- Since $K, \kappa$ are not near 1, different datasets can indeed lead to substantially different returns to scale in terms of reductions in loss. However, under our translations the compute optimal model size is invariant to the training distribution.

- Equation (3) is the only formulation of the underlying scaling law that is compatible with the train-to-train fit given by Equation (8). If we instead used eq. (1), then the transformed scaling law after applying Equation (8) would no longer satisfy the same functional form.

### 5.2 Train-to-test prediction

Next, we want to go beyond the train loss and consider translating the train loss to a test loss for the same model under a different distribution. We now hypothesize that the functional form of the relationship is as

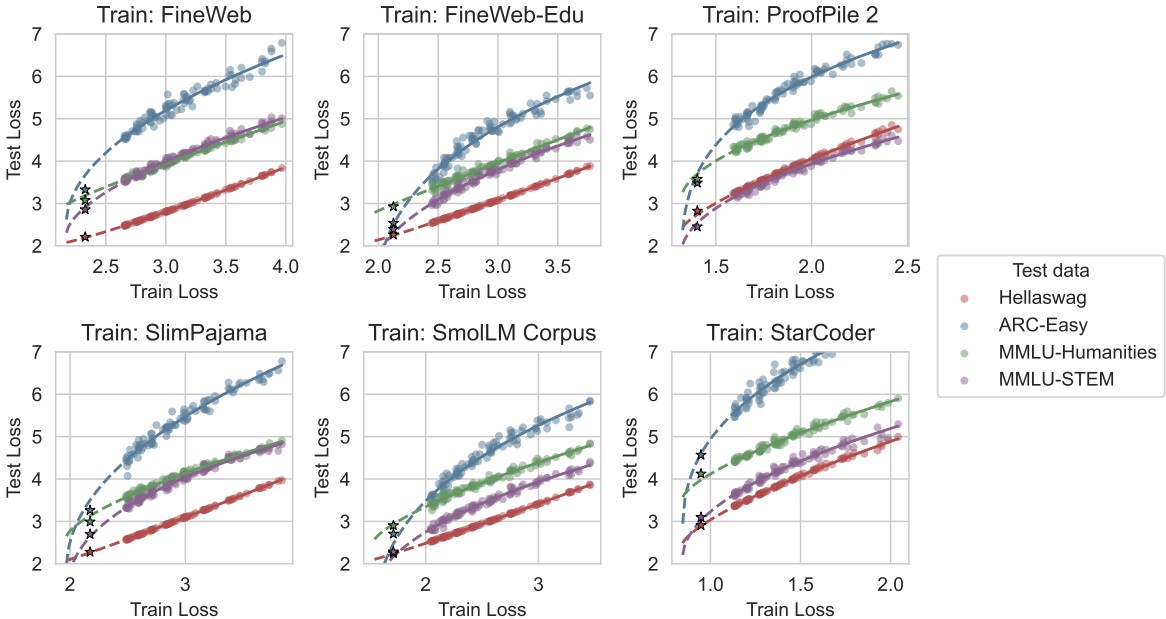

Figure 4: Train-to-test transfer for downstream tasks. On the test set we evaluate the CE loss of the correct multiple choice answer as a cloze task.

follows:

$$L_1(\hat{f}_0^{N,D}) \approx K \cdot \left(L_0(\hat{f}_0^{N,D}) - E_0\right)^{\kappa} + E_{1|0} \tag{9}$$

Note, this is comparing *different* losses, but the *same* model. Further, recall that we define $E_{1|0}$ to be the irreducible error of $L_1$ for the optimal function on $P_0$ with infinite model and data sizes:

$$E_{1|0} := L_1(f_0^*) \tag{10}$$

We can estimate this quantity by fitting a scaling law to $L_1$ under data from $P_0$. In practice, we take the 88 models trained on $P_0$ and evaluate each of them on the OOD test set for $L_1$. This gives a dataset of $(n, d, l)$ tuples that we can use to fit a scaling law and $E_{1|0}$ is the entropy term of that scaling law. Note that this assumes the existence of an underlying scaling law for the test loss that takes the same form as Equation (3).

Results in Figure 3 show predictions to validation sets from the pre-training distributions. Results in Figure 4 translate from train-to-downstream test sets where we use downstream multiple choice questions. Following (Schaeffer et al., 2024b; Madaan et al., 2024), we evaluate the downstream tasks by the cross entropy loss on the correct answer when the question is phrased as a cloze task. Here we show results for Hellaswag (Zellers et al., 2019), ARC-Easy (Clark et al., 2018), and a subset of MMLU (Hendrycks et al., 2020), further results for ARC-Challenge, Openbook QA (Mihaylov et al., 2018), PIQA (Bisk et al., 2020), SciQ (Welbl et al., 2017), Winogrande (Sakaguchi et al., 2021), and the rest of MMLU are in Appendix B.2.

Note that Kaplan et al. (2020) points out a similar trend to Figure 3 in their Section 3.2.2, but they only consider transfer to wikipedia and books and assume the relationship to be linear. By considering a broader array of datasets, we are able to see a more nuanced picture of transfer relationships.

Looking at the train-to-test curves on validation sets in Figure 3, we again see that many of the curves are close to linear ($\kappa$ near 1). However, now there are some notable exceptions when trying to transfer from datasets with little to no code (e.g. FineWeb) to datasets that are entirely code (e.g. StarCoder). These convex curves illustrate diminishing returns to pushing down the FineWeb loss for transfer performance to StarCoder, suggesting that even as we learn a very good model for english it does not improve much on code.

The lines in each plot extend left until we reach the predicted irreducible entropy. Using this fact, another takeaway from Figure 3 is that the asymptotic transfer performance on test sets can be substantially worse

than the performance from training on that dataset directly. This is intuitive, but does imply that including broader training data that includes the test domains we care about is quite important. This is even true for seemingly similar datamixes like SlimPajama and SmolLM. Getting good performance by training on one of the datasets does not imply optimal performance on the other for a given budget.

Turning to downstream tasks in Figure 4 we see substantially higher curvature than we do across pre-training distributions. Moreover, the curves are often concave rather than convex (i.e. $\kappa < 1$). This is interesting since here we are actually seeing increasing returns to improvements in the training loss. We hypothesize that this may occur when due to training dynamics, the target task (like ARC-Easy) lives in some tail of the pre-training distribution that only gets fit by larger models or later in training. Despite this increasing return to scale, we see the improvements in a smooth way because we measure loss rather than accuracy. A detailed discussion of accuracy vs. loss is in Appendix A.

**Implications.** In summary, train-to-test prediction has several implications:

- The predictions across pre-training datasets indicate the importance of data selection. Even if we extrapolate the curves to their ends (where they reach the irreducible error), the loss on transfer datasets do not reach close to the irreducible error for the task, i.e. $E_{1|0}$ does not approach $E_0$.

- Downstream loss is predictable and does not illustrate emergent properties. Tracking this downstream loss gives a smooth proxy to extrapolate performance on tasks of interest.

- Some tasks have convex relationships ($\kappa > 1$) with pre-training loss where decreases in pre-training loss have diminishing returns, while others have concave relationships ($\kappa < 1$) where decreases in pre-training loss have increasing returns. Downstream tasks typically have concave relationships.

### 5.3 Test-to-test prediction

Next, we can move on to test-to-test prediction which can be seen as a composition of the prior two rules. This now involves three different data distributions: $P_0$ the initial training distribution, $P_1$ the target training distribution, and $P_2$ the test distribution that we use to measure loss. Explicitly, we consider:

$$L_2(\hat{f}_1^{N,D}) \approx K \cdot \left( L_2(\hat{f}_0^{N,D}) - E_{2|0} \right)^{\kappa} + E_{2|1} \tag{11}$$

Like train-to-train, these predictions compare the *same* loss on *different* models, but now we are using test loss rather than train loss. In this way, test-to-test can be seen as a generalization of train-to-train. Models are paired when they use the same number of parameters $N$ and number of training tokens $D$.

Results on four downstream losses are shown in Figure 5. Note that now that we are combining three distributions rather than two, there are many more possible combinations. Here we focus on a fixed $P_0$ as FineWeb-edu and show results across training data $P_1$ and test distributions $P_2$. Further results on other sweeps and combinations can be found in Appendix B.3.

Again the fits are usually good and able to extrapolate to models trained with 20x the FLOP budget of the largest one used to fit the curves. The fits are especially good on Hellaswag, but as before the other downstream datasets tend to be substantially noisier. This is magnified now since this evaluation noise affects both the x and y axes when they are both measuring test loss (unlike in train-to-test when only one axis depends on test loss). In the next section we will discuss a practical use case for test-to-test prediction.

## 6 Practical usefulness of loss-to-loss prediction

Consider the following situation that a practitioner could encounter: after having fit a scaling law and performed a large run on one dataset, they want to know what would happen if they trained on a different dataset. They could fit an independent scaling law on the new dataset, but that would not be leveraging the computation that has already been done. Instead, we can use loss-to-loss prediction. This can allow us to get good predictions of the scaling laws and test performance with only a few model runs on the new data distribution since we can leverage information we already have from the original training distribution.

In this section we consider two variants of this situation, one where we fit a scaling law on the new distribution and one where we predict the test loss of training a large model on the new distribution.

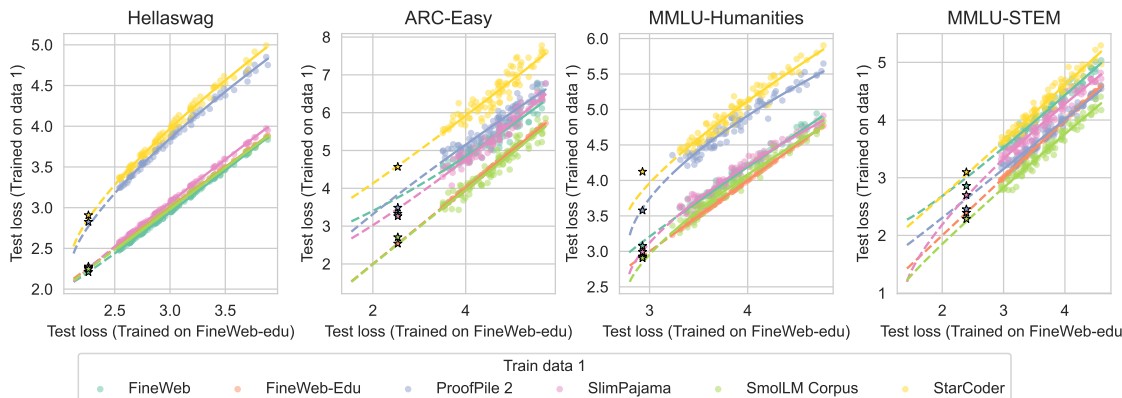

Figure 5: Test-to-test predictions for downstream tasks. Each subplot illustrates a different downstream task. The x-axis always reports the test loss for models trained on FineWeb-edu, and the y-axis shows test loss for all 6 of the different training distributions. Each point represents two models, joined when they share the same model size and training dataset size.

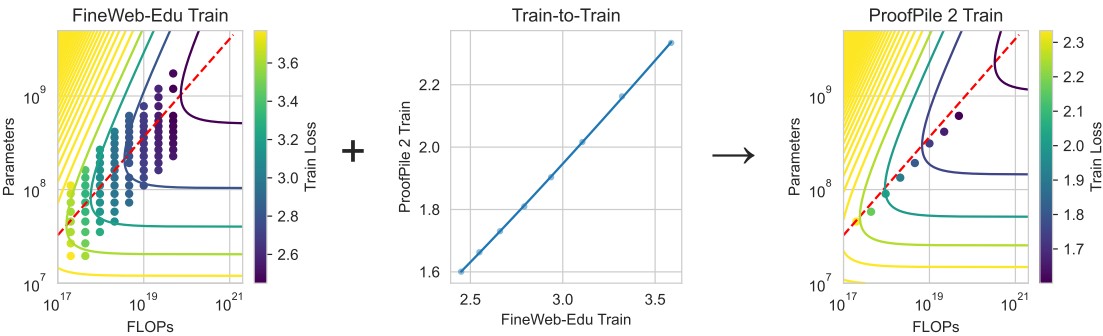

Figure 6: An illustration of using train-to-train prediction to leverage a full set of training runs on FineWeb-edu plus 8 training runs on ProofPile 2 to yield a full scaling law fit on ProofPile 2.

## 6.1 Translating a scaling law

For the scaling law setting, we consider the following scenario. There are two pre-training distributions $P_0$ and $P_1$. Assume that we have already fit a set of 88 small models on $P_0$ so as to fit a scaling law. Then, we fit only 8 small models on a new distribution $P_1$. Note that with our grid, training the full 88 models takes a cumulative 1e21 FLOPs, while only fitting 8 models (one at each FLOP budget) takes only 9e19 FLOPs. We want to get a scaling law on $P_1$.

We will consider two approaches illustrated in Figure 6 and Figure 7:

- (Ours) Train-to-train translation. We fit a train-to-train curve using the 8 models on $P_1$. From this we can translate the scaling law from $P_0$ to $P_1$.[2]

- (Baseline) Independent scaling laws. Here we fit an independent scaling law on $P_1$ from only the 8 models we have that are trained on that dataset.

The point of this experiment is to illustrate how train-to-train fits can unlock an efficient way to fit a new scaling law on a new dataset. Note that as we said above, we should caution that under train-to-train translation the size of the compute optimal model is invariant.

---

[2]Note: while in previous sections, we use $E_1$ or $E_{2|1}$ from the scaling law fits, here we fit any entropy terms that depend $P_1$ as free parameters in the loss-to-loss fits. This is because from small datasets, the scaling law fits are not reliable.

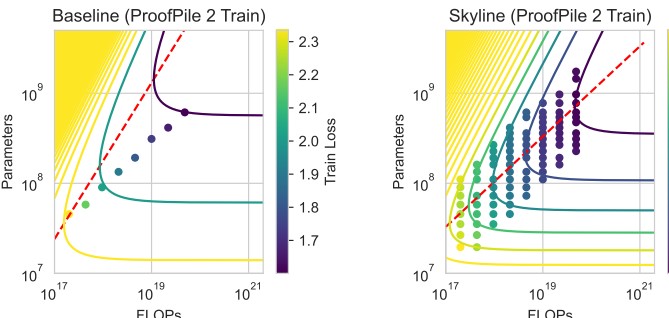

Figure 7: (Left) The baseline just fits the full scaling law on the small dataset of 8 runs on ProofPile 2. (Right) The skyline uses a full suite of models trained on ProofPile 2 to fit a gold-standard scaling law.

We also consider a skyline of fitting a scaling law on $P_1$ with access to all 88 models trained on $P_1$. Then we compute the $R^2$ of each of the three scaling law models (skyline, ours, and baseline) on the entire set of 88 models trained on $P_1$ to assess the goodness of fit. Results are reported in Table 1.

| Target Dataset | Skyline | Ours (mean) | Baseline |
|---|---|---|---|
| FineWeb | 0.992 | **0.990** | 0.961 |
| FineWeb-Edu | 0.992 | **0.990** | 0.953 |
| ProofPile 2 | 0.988 | **0.988** | 0.928 |
| SlimPajama | 0.992 | **0.991** | 0.975 |
| SmolLM Corpus | 0.992 | **0.991** | 0.947 |
| StarCoder | 0.987 | **0.986** | 0.450 |

Table 1: $R^2$ values for scaling laws fit with different methods. For our train-to-train translation we report the mean $R^2$ averaged over the 5 possible values for $P_0$ for each target distribution $P_1$. With only 8 runs from the new dataset, our method can nearly match the skyline which has access to 88 runs from the target dataset. In contrast, the baseline of fitting an independent scaling law fails badly in this limited data regime since it does not leverage prior computation.

We find that loss-to-loss prediction yields substantially better scaling law fits than the baseline. In fact, even with only 8 models on $P_1$, using train-to-train prediction to tranlate the original scaling law nearly matches the $R^2$ of the skyline that has access to all 88 models on $P_1$, up to about 0.001. In contrast, fitting a new scaling law on only this data is very ineffective. This experiment shows that leveraging the existing models from $P_0$ can yield more efficient scaling law fits on a new distribution $P_1$ when using loss-to-loss prediction.

## 6.2 Predicting test loss on a large model

For the test loss setting, we consider the following scenario. There are two pre-training distributions $P_0$ and $P_1$. Assume that we have already fit a set of 4 small models and one larger model (3.3B parameters and 1e21 FLOPs) on $P_0$. Then, we consider a new dataset $P_1$ and fit only 8 small models with various budgets on $P_1$. We want to predict what would happen if we train a large model on $P_1$.

We will consider the approaches illustrated in Figure 8, plus one additional baseline:

- (Ours) General train-to-test prediction. We fit a train-to-test curve across different training sets using the 8 paired small models. Explicitly, we predict $L_2(f_1^{N,D})$ from $L_0(f_0^{N,d})$. This allows us to extrapolate using the train loss of the large model trained on $P_0$ as an input.

- (Ours) Test-to-test prediction. We fit a test-to-test curve using the 8 paired small models. Explicitly, we predict $L_2(f_1^{N,D})$ from $L_2(f_0^{N,d})$. This allows us to extrapolate using the test loss of the large model trained on $P_0$ as an input.

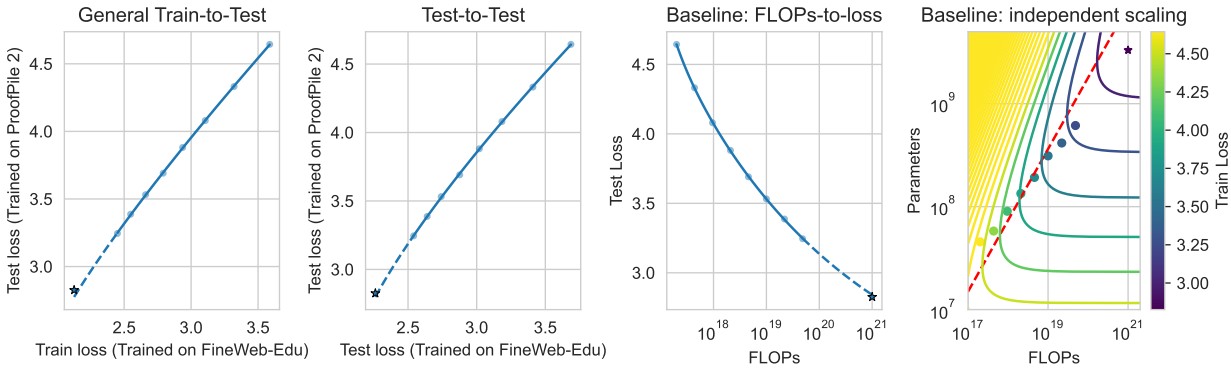

Figure 8: An illustration of four of the methods we consider for making extrapolative predictions of Hellaswag test loss. (Left) General train-to-test prediction uses train loss of models trained on $P_0$ (in this case FineWeb-edu) to predict test loss on models trained on $P_1$ (in this case ProofPile 2). (Center-left) Test-to-test prediction uses test loss of models trained on $P_0$ to predict test loss on models trained on $P_1$. (Center-right) Predicting the test loss only using runs from $P_1$ by fitting the relationship between FLOPs and test loss. (Right) Fitting a full scaling law to $P_1$ using only the limited data from $P_1$.

- (Baseline) FLOPs-to-test. As a first baseline that does not use information from $P_0$, we can fit a curve from FLOPs to test loss. Since each of the models is near the chinchilla-optimal model size for the FLOP budget, it is reasonable to fit a curve and extrapolate it here.

- (Baseline) Independent scaling law. As before, we can fit a full scaling law to the set of small models on $P_1$ and extrapolate the predictions.

- (Baseline) Identity. As an even simpler baseline, we can just predict that the test loss when training on $P_1$ is exactly the same as training on $P_0$.

| Target Loss | General Train-to-Test | Test-to-Test | FLOPs-to-loss | Scaling law | Identity |
|---|---|---|---|---|---|
| Hellaswag | 1.6% | **1.2**% | 1.7% | 2.1% | 9.2% |
| ARC-Easy | **10.2**% | 17.6% | 14.3% | 16.8% | 24.8% |
| MMLU-Humanities | **2.8**% | 23.1% | 4.4% | 4.7% | 11.0% |
| MMLU-STEM | 6.4% | 6.4% | **5.9**% | 7.6% | 11.5% |

Table 2: Relative error, i.e. $\frac{|\text{pred}-\text{actual}|}{\text{actual}}$, of various methods for predicting the test loss of the the extrapolation run trained on a new dataset. All runs assume that we have already run a set of pre-training runs on FineWeb-edu as $P_0$. All values are averaged across the 5 possible target pre-training datasets $P_1$. Loss-to-loss predictions are usually the most accurate.

We report results in terms of the relative error ($\frac{|\text{pred}-\text{actual}|}{\text{actual}}$) of the prediction of the test loss for various test sets in Table 2. We find that the loss-to-loss methods tend to perform the best. This makes sense because we are able to leverage extra information, especially the loss of the large model on $P_0$ to improve the predictions. The baselines have no way to incorporate this information that we know from already having trained models on $P_0$. Note that train-to-test tends to out-perform test-to-test on the noisier eval datasets (i.e. those other than Hellaswag). This makes sense because using a noisy $x$ variable to regress onto a noisy $y$ variable is going to be higher variance than using a lower variance $x$ variable. Especially since standard train-to-test prediction suggests that there is no more information in the test loss on $P_0$ compared to the train loss. An interesting direction for future work is to figure out how to leverage this type of prediction to perform data selection.

# 7 Discussion

Here we discuss the takeaways of our findings, some limitations, and directions for future work.

**Takeaways.**

- Loss-to-loss fits with shifted power laws provide a good description of empirical trends across a variety of pre-training datasets and to downstream tasks. These fits can effectively extrapolate well beyond the scale they were trained on.

- Loss-to-loss prediction is of scientific interest since it provides several insights into the nature of how training data affects models and how transfer performance scales predictably.

- Loss-to-loss predictions can be practically valuable for translating scaling laws and predicting test loss of large models trained on new data.

**Limitations and caveats.**

- Our fits rely on estimating the asymptotic entropy of various scaling laws. This is a fundamentally difficult quantity to estimate and we hypothesize that where our fits fail it is often due to poor estimates of this quantity. Moreover, we hypothesize that when our fits fail to extrapolate beyond the 20x results reported in the paper, it is likely due to errors in estimating these irreducible loss terms.

- Note that many of the train-to-test and test-to-test fits have noisier trends, especially at high losses. It is not totally clear if this is pure noise or may be indicative that the power law trend does not hold as globally as we hypothesize. Future work could dive into this issue more directly.

- We only test on a relatively small set of downstream tasks compared to all possible choices. We also focus on multiple choice tasks instead of generative tasks since they have been more extensively studied in prior work and have easier to compute proxy loss metrics.

- Our results hold for our specific choices of hyperparameters and may not hold under some other choices. In particular, we would be interested in checking robustness to pre-training hyperparameters like sequence length, batch size, and learning rate.

**Future work.**

- One exciting direction is to take the implications of the loss-to-loss relationships further so as to directly inform data mixing and filtering. Once we have reliable predictions, we can use those to inform choices about which data to train on. Perhaps this could use the scaling laws derived here in combination with recent relating scaling laws and data mixtures (Jiang et al., 2024; Ye et al., 2024).

- We hope to gain a tighter theoretical understanding as to why the loss-to-loss relationships are so clean by studying simplified models. In Appendix G we attempt to connect some of the existing theory literature to our results, and show that a prototypical version of train-to-train transfer emerges in a class of previously studied linear models. It would be interesting have a better theoretical understanding of train-to-test transfer as well as a richer model that could capture the full extent of the phenomena that we observe in practice.

- Our results connect surprisingly disparate datasets. We are able to predict performance on code data from data that contains no code and visa-versa. It would be nice to have a better mechanistic understanding of how this works. It is possible that all the models converge to "features" that share some high level distributional properties (e.g. similar eigenvalue decay of the covariance). Or at a different level of granularity, it is possible that there the data is more similar than we think and there is a large enough amount of English in code and visa versa that losses are predictive. Or perhaps there are particular shared structures that emerge, e.g. in context learning.

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

## A  From loss to accuracy

### A.1  Train-to-error

We focus on loss-to-loss prediction, but it of course would be useful to be able to predict accuracy. Prior work (Schaeffer et al., 2024a;b; Du et al., 2024) indicates that predicting accuracy from loss can be difficult, and we generally agree. However, other work (Gadre et al., 2024) claims that downstream accuracy can be predictable in some cases and we want to consider here whether accuracy is predictable in our data with methods similar to those presented in the main text.

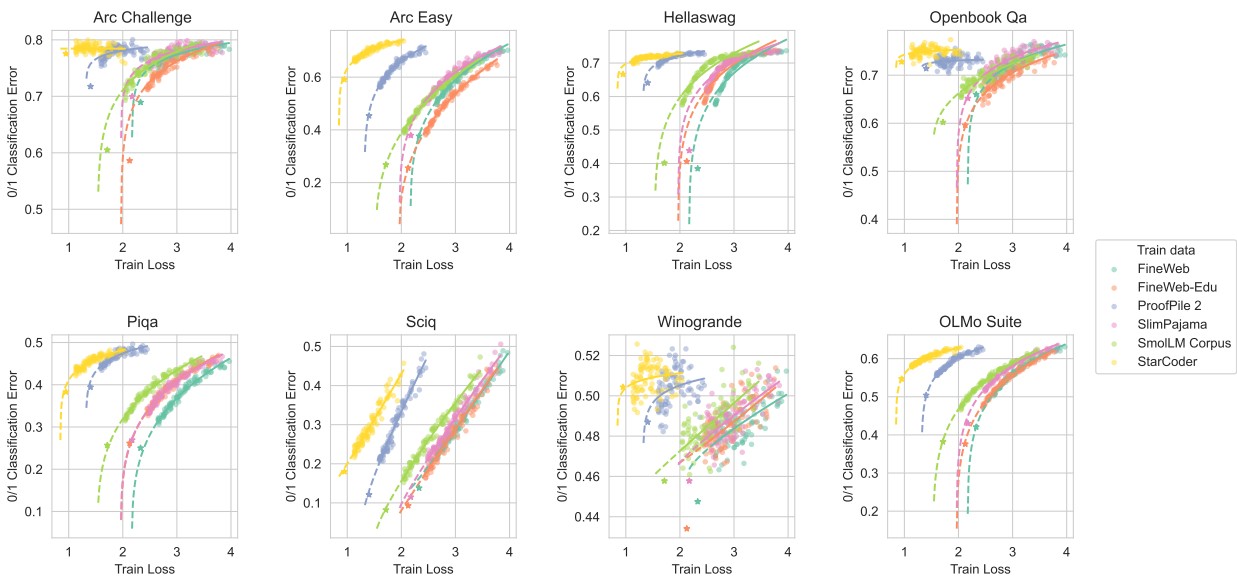

Figure 9: Fitting training loss to accuracy on the OLMo tasks individually (first 7 subplots), and then in aggregate (bottom right). Unlike the plots in the main paper where each line only fits 2 parameters $K, \kappa$, here we also fit a third parameter in place of $E_{1|0}$.

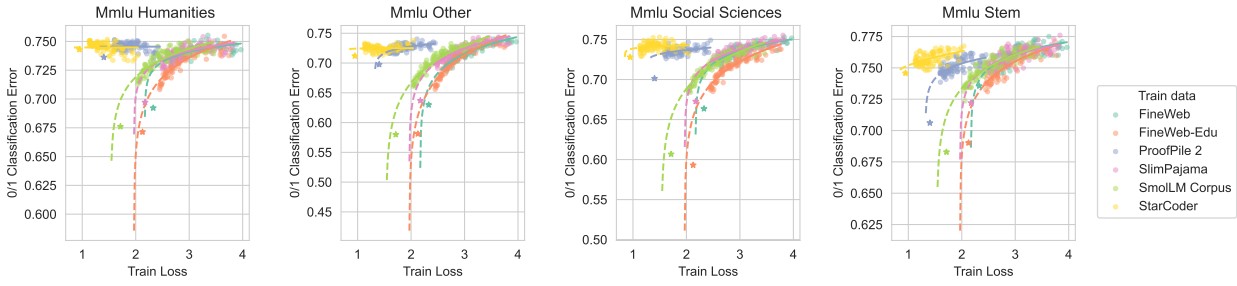

Figure 10: Fitting training loss to accuracy on MMLU splits.

In particular, (Gadre et al., 2024) specifically finds that when they select a subset of 17 particularly easy benchmarks (where performance is better than chance for small models), then they can get good predictions for the average accuracy by fitting shifted power laws with a methodology similar to the one that we use for loss-to-loss prediction (but where $E_{1|0}$ is treated as a free parameter). We are able to reproduce a similar result on our suite of 7 tasks from OLMo, see Figure 9. Explicitly, we fit the following relationship to and let the multiple choice error $\mathcal{E}_1$ (i.e. 1 - accuracy):

$$\mathcal{E}_1(\hat{f}_0^{N,D}) \approx K \cdot \left( L_0(\hat{f}_0^{N,D}) - E_0 \right)^{\kappa} + M \tag{12}$$

where $Err$ is the error and unlike in the main text we are now fitting 3 parameters $K, \kappa, M$ instead of just $K, \kappa$.

The fits are fairly good for the aggregate, but it is clear that some of the fits (e.g. Hellaswag and ARC challenge) are systematically wrong. They end up overestimating the error because power law fits fundamentally cannot handle the fact that bad models will perform at random chance. The asymptotics of a power law mean that as $L \to \infty$ we get $Err \to \infty$, which is not possible. This is fundamentally related to the loss perspective on emergence (Du et al., 2024) where for multiple choice tasks there is some value of loss where the models start performing better than random chance. This is also perhaps even more clear for MMLU in Figure 10. In general, we would not expect this technique to work on individual tasks and especially not on more challenging tasks.

One potential remedy for this issue would be to introduce a fourth parameter to the fits that can handle the transition from predicting at chance to making progress. Explicitly, we can let the curve be the soft-min $(\mathrm{softmin}(x, y) = -\log(\exp(-\alpha x) + \exp(-\alpha y))$ for $\alpha = 10)$ between a constant $c$ representing the chance error rate and the shifted power law from before. Explicitly:

$$\mathcal{E}_1(\hat{f}_0^{N,D}) \approx \mathrm{softmin}(c, K \cdot \left(L_0(\hat{f}_0^{N,D}) - E_0\right)^\kappa + M) \tag{13}$$

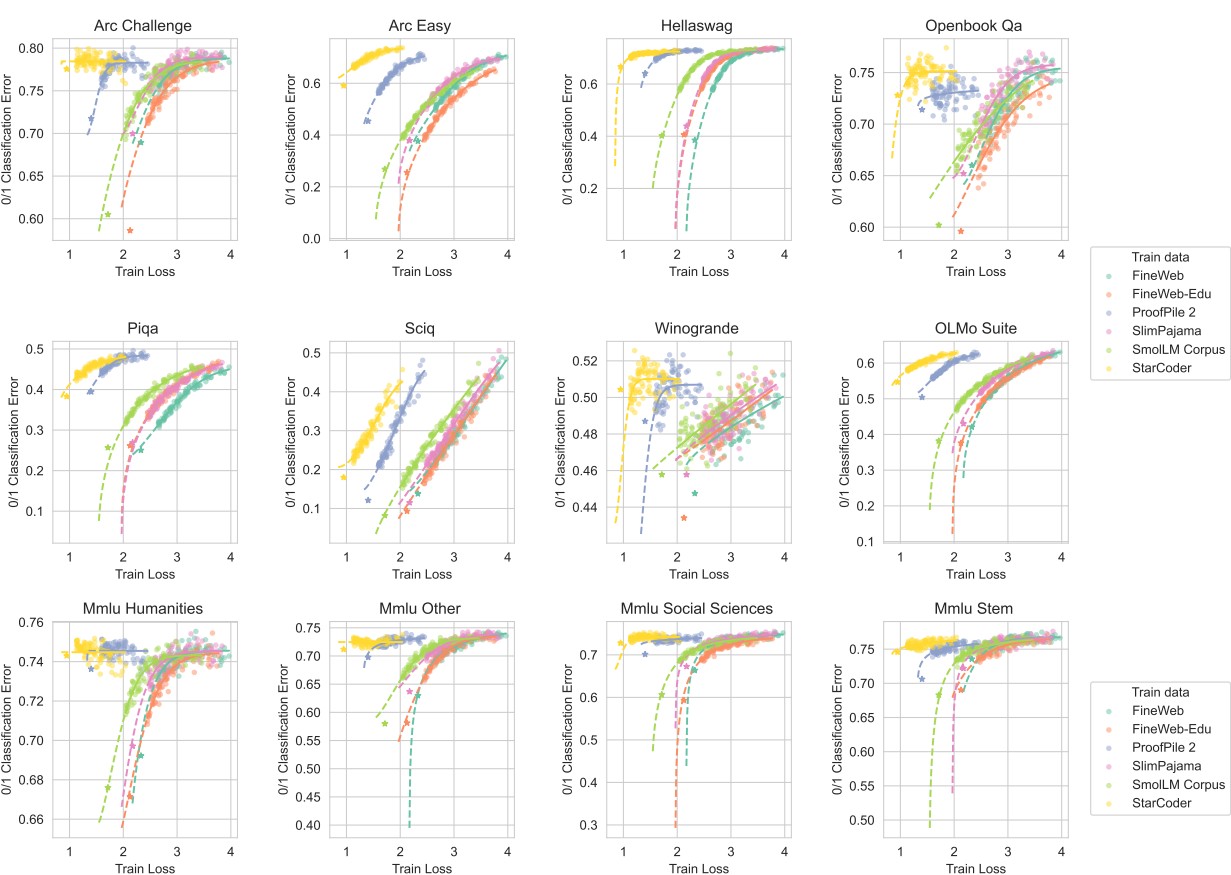

Figure 11: The relationship between downstream CE loss and classification error shows unified trends across pre-training distributions, i.e. it seems that all points roughly fit onto one trend line regardless of their color.

Results for this approach with 4 learned parameters per curve are shown in Figure 11. In general, we find that this seems to help (e.g. on Hellaswag and MMLU), but may introduce bias on others (e.g. on Openbook or SciQ). We think this is a promising approach and does seem to yield more robust predictions than the prior approach on harder tasks like MMLU. But, we are not certain that these fits are quite right or as

universal as the simple shifted power laws relating losses. As seen in prior work, computing accuracies is nuanced since it is a hard metric that also depends on the wrong answers (Schaeffer et al., 2024b). As such, we focus the main paper on losses which we find to more consistently obey shifted power law relationships.

### A.2  Test-to-error

For similar reasons, we also found it difficult to fit loss-to-error maps from the downstream CE loss to the classification error. However, while the exact functional form of the dependence is unclear, there is useful information in the loss-to-error plots in Figure 12. Importantly, there is convergence across pre-training distributions where irrespective of the pre-training distribution there is a relatively consistent relationship between downstream CE loss and classification error. This is markedly different from the patter we see when looking at train loss where each pre-training dataset yields a different relationship between train and any test loss or error.

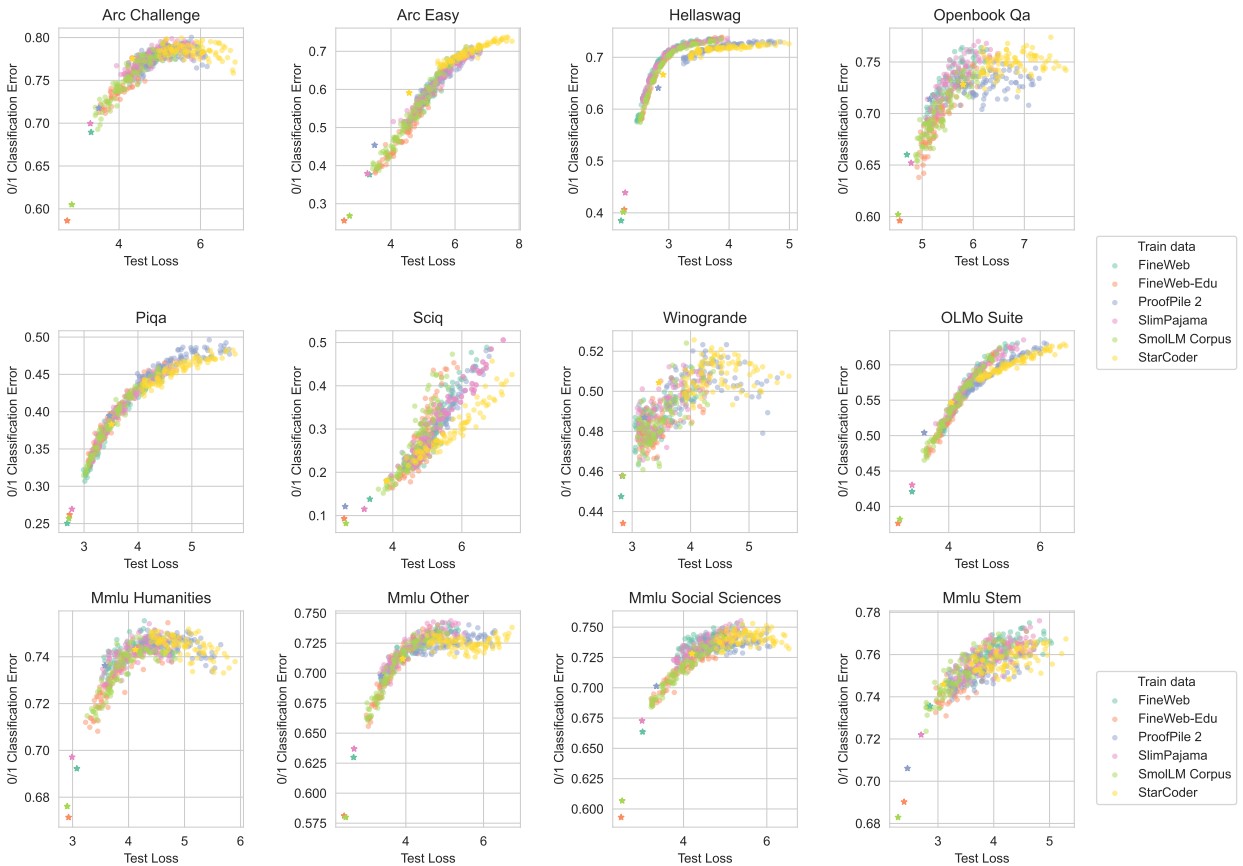

Figure 12: The relationship between downstream CE loss and classification error shows unified trends across pre-training distributions, i.e. it seems that all points roughly fit onto one trend line regardless of their color.

The fact that the trend from test loss to error is unified across pre-training data suggests that this test loss is a good proxy measure for the downstream task and supports using it as our main endpoint in the paper. In particular, if we consider the causal relationships between different variables, we are suggesting that the train loss only causes the downstream accuracy through a mediating variable that is the downstream CE loss on the correct answer. As a result, once we compute the downstream CE loss, we break the causal relationship between pre-training data and downstream accuracy. This seems to be generally true, but may not be strictly true at high loss values (e.g. on SciQ or Hellaswag). But, this does suggest that the CE error is a useful proxy since it mediates the pre-training-specific effects from the test accuracy.

# B  Additional loss-to-loss results

## B.1  $R^2$ and extrapolation error for train-to-train

For the train-to-train fits in Figure 2 we provide tables here with the $R^2$ value of the fits on the training data and the relative error of the predictions on the extrapolation point. Note that the gap between our method and linear fits is much larger for the train-to-test and test-to-test, so we focus on this case where the comparison is closest.

Both methods fit two parameters: $K, \kappa$ for ours and a slope and intercept for the linear method. The results show that the $R^2$ values are quite close, but the extrapolations are generally better for our method. This is because our method respects the differences in asymptotic entropy between different datasets, while the linear fits cannot do this. The gap is especially large on the most different dataset, StarCoder, where the nonlinearity of the relationship is most clear.

| Target data | Ours | Linear |
|---|---|---|
| FineWeb | **0.9998** | **0.9998** |
| ProofPile 2 | **0.9990** | 0.9989 |
| SlimPajama | **0.9997** | 0.9996 |
| SmolLM Corpus | **0.9999** | **0.9999** |
| StarCoder | **0.9979** | 0.9972 |

Table 3: Train-to-train $R^2$ when the source data is FineWeb-edu (other datasets are similar).

| Target data | Ours | Linear |
|---|---|---|
| FineWeb | **0.141%** | 0.300% |
| ProofPile 2 | **0.086%** | 1.429% |
| SlimPajama | 1.339% | **0.643%** |
| SmolLM Corpus | **0.649%** | 0.822% |
| StarCoder | **1.957%** | 5.795% |

Table 4: Train-to-train relative error of the predictions at the extrapolation point when the source data is FineWeb-edu (other datasets are similar).

## B.2    Train-to-test

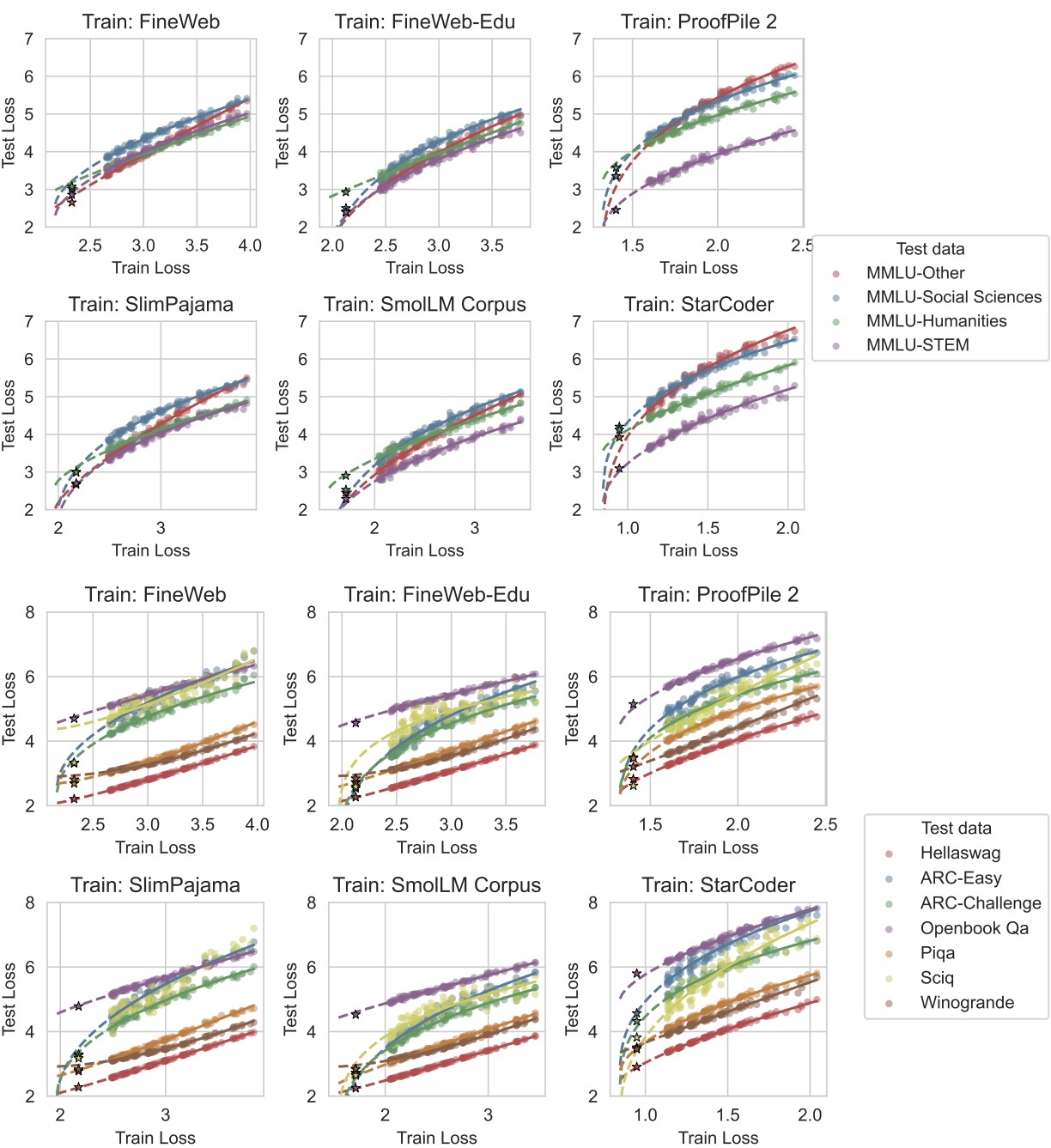

Figure 13: Train-to-test predictions across all individual downstream tasks.

### B.3 Test-to-test

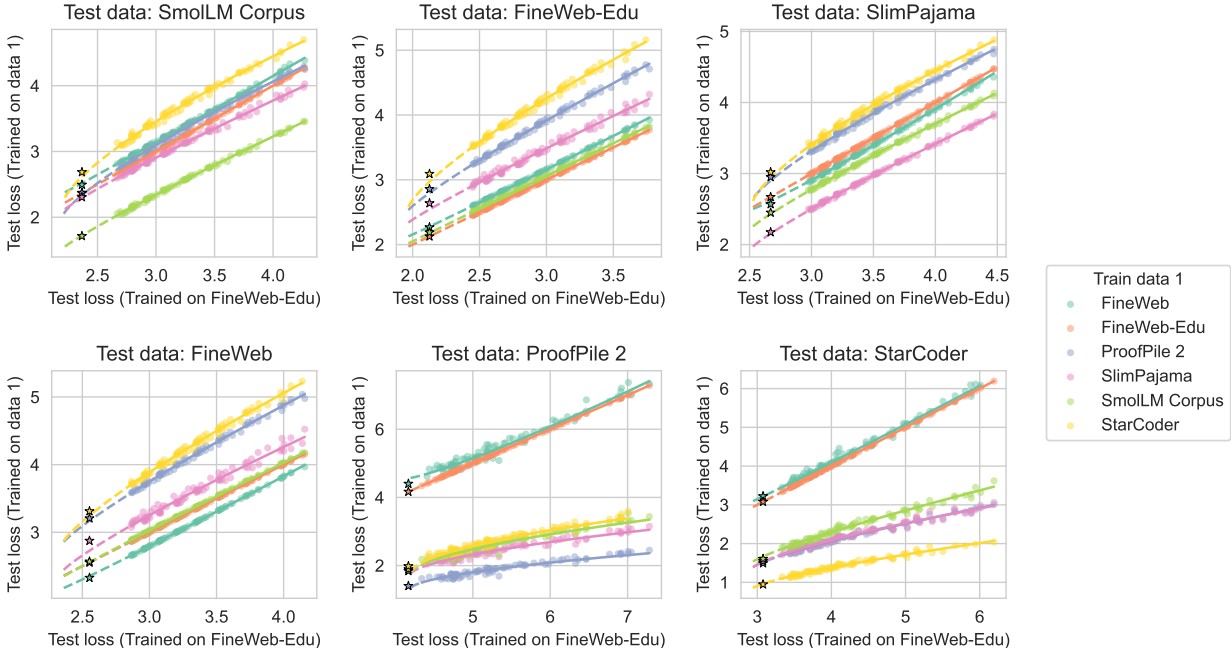

Figure 14: Test-to-test prediction using the validation sets from pre-training data as the targets. Each subplot shows a different test loss. Within each subplot, the training data $P_0$ is always FineWeb-Edu and the curves illustrate all of the 6 possible options for $P_1$. Each point corresponds to two models.

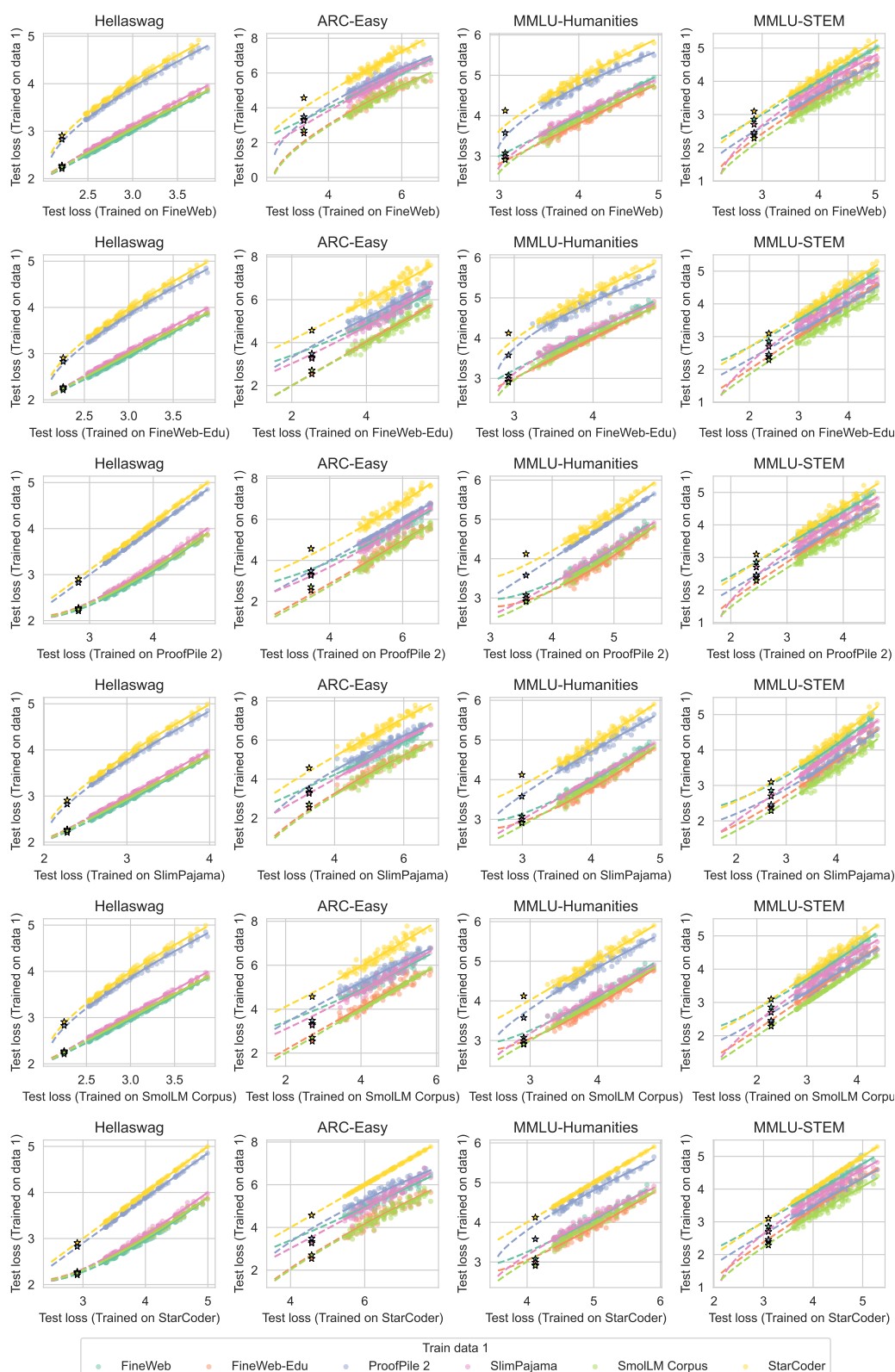

Figure 15: Test-to-test prediction on the four losses from the main text. Each row shows a different training loss $P_0$ on the x-axis. Each point corresponds to two models.

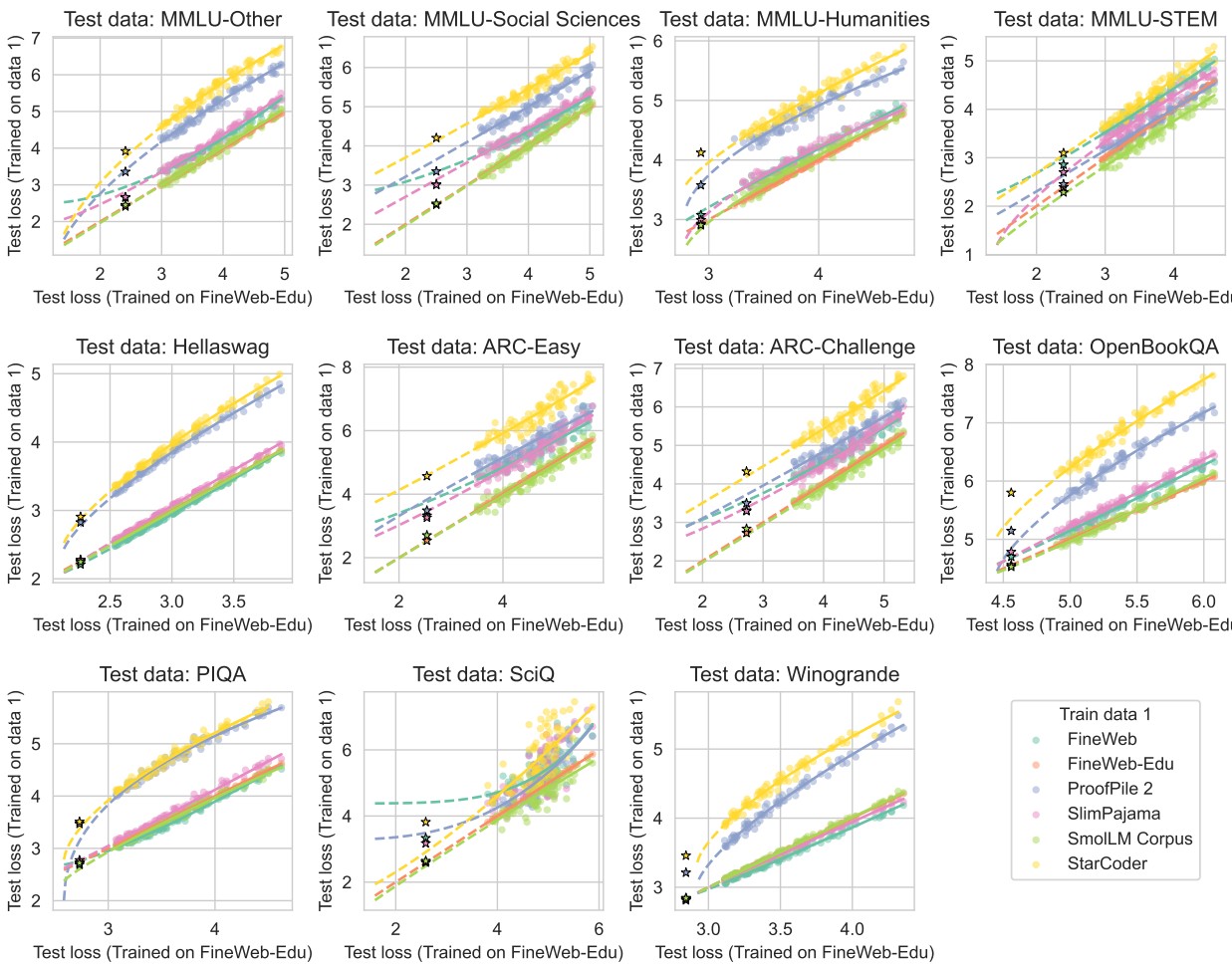

Figure 16: Test-to-test prediction on all 11 downstream losses we consider. The training data $P_0$ is fixed to FineWeb-edu in all subplots. Each point corresponds to two models.

## C   Scaling law fits

We follow the methodology from Hoffmann et al. (2022); Besiroglu et al. (2024) for fitting scaling law curves and illustrate fits for both Equation (3) and Equation (1). In particular, we have on average 88 datapoints for each training dataset, each consisting of the model size, number of training tokens, and final loss. We use the estimate that FLOPs is $6ND$ when reporting FLOPs. We then fit the curves using L-BFGS in log space with Huber loss. We plot the results and give the parameter values.

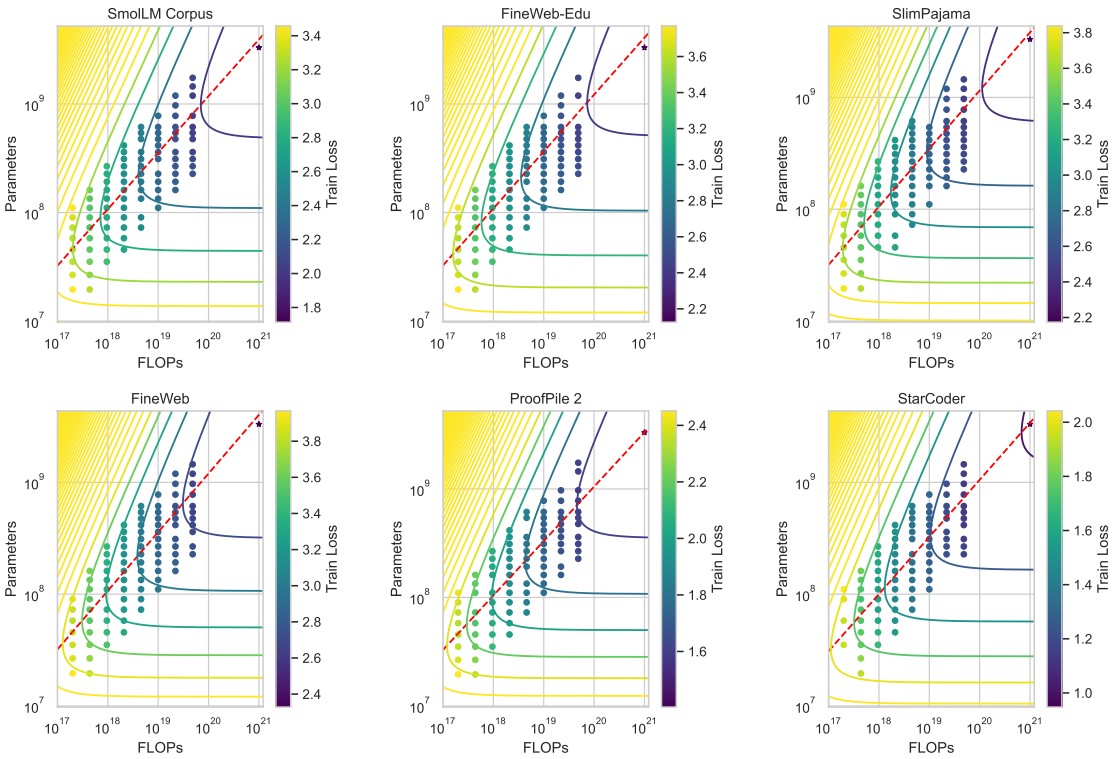

Figure 17: Contour plots for the curves fit with Equation (3) (our version of the scaling law parameterization). Red line indicates the optimal model size. The star point is not used for fitting the curves.

| Data | $A$ | $B$ | $E$ | $\alpha$ | $\beta$ | $a$ |
|---|---|---|---|---|---|---|
| SmolLM Corpus | 7.79e+07 | 1.06e+09 | 1.53 | 0.42 | 0.45 | 0.52 |
| FineWeb-Edu | 6.68e+07 | 8.90e+08 | 1.97 | 0.41 | 0.46 | 0.52 |
| SlimPajama | 7.47e+07 | 1.06e+09 | 1.97 | 0.40 | 0.43 | 0.52 |
| FineWeb | 6.79e+07 | 9.31e+08 | 2.17 | 0.41 | 0.45 | 0.52 |
| ProofPile 2 | 2.14e+07 | 3.29e+08 | 1.32 | 0.45 | 0.46 | 0.50 |
| StarCoder | 2.23e+07 | 3.78e+08 | 0.85 | 0.45 | 0.47 | 0.51 |

Table 5: Parameters for the curves fit with Equation (3) (our version of the scaling law parameterization). $a = \frac{\beta}{\alpha+\beta}$ is the exponent of the optimal model size relative to FLOPs.

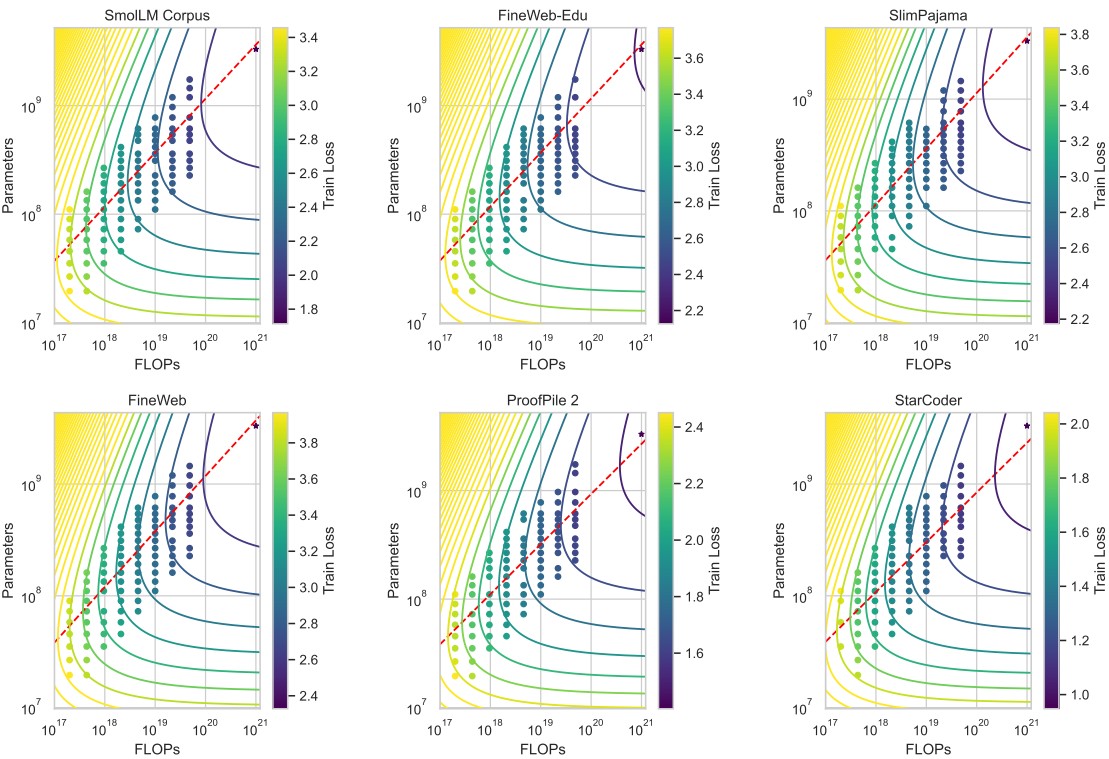

Figure 18: Contour plots for the curves fit with Equation (1) (the chinchilla version of the scaling law parameterization). Red line indicates the optimal model size. The star point is not used for fitting the curves.

| Data | $A$ | $B$ | $E$ | $\alpha$ | $\beta$ | $a$ |
|------|-----|-----|-----|----------|---------|-----|
| SmolLM Corpus | 2.44e+03 | 6.92e+03 | 1.55 | 0.45 | 0.44 | 0.50 |
| FineWeb-Edu | 2.52e+03 | 7.16e+03 | 2.00 | 0.45 | 0.45 | 0.50 |
| SlimPajama | 2.05e+03 | 6.02e+03 | 2.01 | 0.44 | 0.44 | 0.50 |
| FineWeb | 1.64e+03 | 4.20e+03 | 2.15 | 0.43 | 0.42 | 0.50 |
| ProofPile 2 | 3.77e+03 | 3.59e+03 | 1.33 | 0.51 | 0.43 | 0.46 |
| StarCoder | 7.75e+03 | 4.19e+03 | 0.86 | 0.55 | 0.44 | 0.45 |

Table 6: Parameters for the curves fit with Equation (1) (the chinchilla version of the scaling law parameterization). $a = \frac{\beta}{\alpha+\beta}$ is the exponent of the optimal model size relative to FLOPs.

# D    Hyperparameters

Here we list all the hyperparameters for our model training. We always report the final losses at the end of training and use the approximation that FLOPs is $6ND$ to compute flops from model size and data size. Each training run is conducted using the OLMo (Groeneveld et al., 2024) codebase with the listed hyperparamters. We construct grids of various model sizes for each FLOP budget, as shown in the previous section.

Table 7: Model parameters (Groeneveld et al., 2024; Wortsman et al., 2023; Zhao et al., 2024)

| Parameter | Value |
|---|---|
| $n$ | 6-24 for small models, 40 for the 3.3B model |
| Number of heads | $n$ |
| Head dimension | 64 |
| MLP hidden multiplier | 4 |
| Depth | $n$ |
| Context length | 512 |
| Activation | GeLU |
| Positional encoding | RoPE |
| Biases | False |
| Normalization | PyTorch Layernorm |
| QK normalization | True |
| Precision | Mixed, bfloat16 |
| Tokenizer | Llama2 |

Table 8: Training parameters (Groeneveld et al., 2024; Wortsman et al., 2023; Zhao et al., 2024)

| Parameter | Value |
|---|---|
| Optimizer | Adam |
| Batch size | 1024 |
| Learning rate | 1e-3 |
| Schedule | Linear warmup, cosine decay |
| Warmup steps | 20% of total steps |
| z-loss coefficient | 1e-4 |
| Weight decay | 0.0 |
| $\beta_1$ | 0.9 |
| $\beta_2$ | 0.95 |
| $\epsilon$ | 1e-15 |

# E    Full loss-to-loss parameter fits from Figure 1

Table 9: Train-to-train fits

| Data 0 | Data 1 | $\kappa$ | $K$ | $E_0$ | $E_1$ |
|---|---|---|---|---|---|
| FineWeb-Edu | FineWeb | 1.00 | 1.01 | 1.97 | 2.17 |
| FineWeb-Edu | FineWeb-Edu | 1.00 | 1.00 | 1.97 | 1.97 |
| FineWeb-Edu | ProofPile 2 | 1.07 | 0.60 | 1.97 | 1.32 |
| FineWeb-Edu | SlimPajama | 0.97 | 1.05 | 1.97 | 1.97 |
| FineWeb-Edu | SmolLM Corpus | 1.01 | 1.07 | 1.97 | 1.53 |
| FineWeb-Edu | StarCoder | 1.10 | 0.63 | 1.97 | 0.85 |

Table 10: Test-to-test fits

| Train data 0 | Train data 1 | $\kappa$ | $K$ | $E_{2\|0}$ | $E_{2\|1}$ |
|---|---|---|---|---|---|
| FineWeb-Edu | FineWeb | 1.05 | 0.98 | 2.12 | 2.08 |
| FineWeb-Edu | FineWeb-Edu | 1.00 | 1.00 | 2.12 | 2.12 |
| FineWeb-Edu | ProofPile 2 | 0.74 | 1.60 | 2.12 | 2.39 |
| FineWeb-Edu | SlimPajama | 0.95 | 1.11 | 2.12 | 2.08 |
| FineWeb-Edu | SmolLM Corpus | 0.99 | 1.01 | 2.12 | 2.10 |
| FineWeb-Edu | StarCoder | 0.74 | 1.64 | 2.12 | 2.48 |

Table 11: Train-to-test fits

| Train data | Test data | $\kappa$ | $K$ | $E_0$ | $E_{1\|0}$ |
|---|---|---|---|---|---|
| FineWeb-Edu | Hellaswag | 1.08 | 0.93 | 1.97 | 2.12 |
| FineWeb-Edu | ARC-Easy | 0.36 | 4.68 | 1.97 | 0.07 |
| FineWeb-Edu | MMLU-Humanities | 0.96 | 1.14 | 1.97 | 2.79 |
| FineWeb-Edu | MMLU-STEM | 0.53 | 2.35 | 1.97 | 1.41 |

## F  Loss-to-loss procedure

Here we summarize the procedure used to produce the loss-to-loss curves in the paper. We present this in the general form, assuming we want to predict $L_i(\hat{f}_j^{N,D})$ given $L_k(\hat{f}_\ell^{N,D})$.

1. We independently fit scaling laws to $L_i(\hat{f}_j^{N,D})$ and $L_k(\hat{f}_\ell^{N,D})$ on the full datasets using the method described in Appendix C. We only use these fits to extract the irreducible entropy terms $E_{i|j}$ and $E_{k|\ell}$.

2. Then we fit $K, \kappa$ by linear regression in log space after shifting the data by $E_{i|j}$ and $E_{k|\ell}$.

In Section 6 we also consider fits from smaller datasets where $E_{i|j}$ is difficult to fit well given the limited sample size. In these cases, we instead let $E_{i|j}$ be a free parameter and fit all of $K, \kappa, E_{i|j}$ via non-linear least squares optimization using scipy.

## G  Comment on theoretical implications

There is now a growing body of literature on the theory of loss scaling in large neural networks (see, e.g., Bahri et al. (2024); Lin et al. (2024); Sharma and Kaplan (2022); Maloney et al. (2022); Canatar et al. (2021); Dohmatob et al. (2024); Hutter (2021); Wei et al. (2022); Michaud et al. (2023); Jain et al. (2024); Bordelon et al. (2020); Atanasov et al. (2024); Nam et al. (2024); Bordelon et al. (2024b); Paquette et al. (2024) and references therein). For example, Lin et al. (2024) derives an expression for the loss scaling at finite model size and dataset size in a sketched linear model and single-pass SGD setting. Bahri et al. (2024) and Atanasov et al. (2024) considered a similar problem in an analogous student-teacher network setting, but in the asymptotic regimes where either the dataset size or model size was taken to infinity.

However, there is comparatively less theoretical work on understanding the effects of the data distribution on the scaling laws, and on disentangling the two different types of scaling laws in Equation (1) and Equation (3). This is partially because in the asymptotic regime when $N \to \infty$ or $D \to \infty$, both forms given rise to the same scaling in the other variable and because empirically both result in "reasonable" fits to the data. Works like Lin et al. (2024) derive bounds which include cross terms involving both $N$ and $D$, but it remains unclear if these cross terms can be interpreted as those coming from the polynomial form of Equation (3).

In this work, we use the scaling law in Equation (3) since it yields valid scaling law translations (though our results do not necessarily rule out other parametrizations). This leads us to ask if existing theoretical models prefer the functional form of Equation (3) versus, e.g., Equation (1). In this section, we consider this

question in a simple linear model that has been considered in many previous works to theorize about scaling laws (Bordelon et al., 2020; Maloney et al., 2022; Lin et al., 2024). Our goal here is not to derive a novel result, but rather to show that a simplified version of the train-to-train (in-domain) loss transfer emerges in the existing theory, and that the scaling law is qualitatively described by an equation that is roughly analogous to Equation (2). However, we defer analysis of the train-to-test and test-to-test setting for future work, which, to the best of our knowledge, is not captured by any theoretical model studied in the literature.

### G.1 Generalized linear model

As in Canatar et al. (2021); Spigler et al. (2019); Cui et al. (2021); Maloney et al. (2022), we consider data $x_i$ that has $M$ features whose covariance has a spectrum that exhibits the empirically-motivated power-law behavior

$$\lambda_i = \frac{1}{i^{\beta+1}}, \quad i : 1, \dots, M. \tag{14}$$

It is straightforward to construct a features dataset that satisfies this property. For example, for any random orthogonal matrix $O$ we can construct a dataset of dimension $D$-by-$M$ by taking the covariance to be $\Sigma = O\Lambda O^\top$ with $\Lambda = \text{diag}(\lambda_i)$, and sample $D$ samples from $\mathcal{N}(0, \Sigma)$.

To avoid directly working in the large feature space, these features are projected down into a smaller set (this controls the extent to which the learner can resolve the features). Mechanically, we also want to disentangle the size of the dataset $D$ from the number of parameters of our model $N$. This can be achieved through a linear map

$$\phi_\text{a}(x) = \sum_{i=1}^{M} v_{\text{a}i}\, x_i, \quad \text{a} : 1, \dots, N. \tag{15}$$

The weights are drawn from a normal distribution $v_{\text{a}i} \sim \mathcal{N}(0, \sigma_v^2 M^{-1})$. The learned model is

$$f(x; \theta) = \sum_{\text{a}=1}^{N} \theta_\text{a} \phi_\text{a}(x), \tag{16}$$

where $\theta$ are the model parameters, and for simplicity we have assumed a scalar output (i.e. a single label per sample). The labels are given by

$$y = \sum_{i=1}^{M} w_i x_i, \tag{17}$$

where $w_i \sim \mathcal{N}(0, \sigma_w^2)$. We optimize the squared loss

$$\mathcal{L}(\theta) = \frac{1}{2} \left\| f(x; \theta) - y \right\|^2. \tag{18}$$

Note that for simplicity we do not consider the ridge term (we will work far enough into the underparametrized regime $N < D$, where the ridge term does not significantly contribute to the loss) and work in the limit of zero label noise. The analytic solution for the optimal parameters $\theta^*$ is straightforward to compute and given by (Maloney et al., 2022; Atanasov et al., 2024)

$$\theta^* = y^\top \phi (\phi^\top \phi)^{-1}. \tag{19}$$

Since there exists an exact formula for the optimal parameters, this can be seen as effectively performing infinite passes on the training data.

Once we obtain a set of optimal parameters $\theta^*$ we evaluate the loss on a large held-out validation set whose samples $\hat{x}$ are also drawn from $\mathcal{N}(0, \Sigma)$:

$$\begin{aligned}
\hat{\mathcal{L}}(\theta^*) &= \frac{1}{2} \mathbb{E}_{\hat{x} \sim \mathcal{N}(0, \Sigma)} \left\| f(\hat{x}; \theta^*) - \hat{y} \right\|^2 \\
&= \frac{1}{2} \mathbb{E}_{\hat{x} \sim \mathcal{N}(0, \Sigma)} \left\| f(\hat{x}; \theta^*) - \hat{x} w^\top \right\|^2.
\end{aligned} \tag{20}$$

The number of features is larger than the number of parameters and dataset size $M \gg N, D$, such that the loss on the validation set decreases as the size of the train set is made larger. The expectation can be evaluated in closed form and is given by (Maloney et al., 2022)

$$\hat{\mathcal{L}}(\theta^*) \equiv \hat{\mathcal{L}}(N, M, D) = \frac{\sigma_w^2}{2}\left(\frac{\Delta}{1 - N/D}\right), \tag{21}$$

where the quantity $\Delta$ satisfies the trace equation

$$1 = \text{tr}\left[\Sigma\left(\Delta\mathbf{1}_M + N\Sigma\right)^{-1}\right]. \tag{22}$$

In the eigenbasis, we can write this as

$$1 = \sum_i \frac{\lambda_i}{\Delta + N\lambda_i}. \tag{23}$$

Plugging in Equation (14) for our eigenvalue scaling, we therefore have

$$1 = \sum_{i=1}^{M} \frac{1}{\Delta i^{\beta+1} + N}. \tag{24}$$

When the spectrum is dense ($M \to \infty$) we can approximate this as[3]

$$1 \approx \int_1^\infty \frac{dz}{\Delta z^{\beta+1} + N} = \frac{1}{\beta\Delta}{}_2F_1\left(1, \frac{\beta}{\beta+1}, 2 - \frac{1}{\beta+1}, -\frac{N}{\Delta}\right), \tag{25}$$

where ${}_2F_1$ is the hypergeometric function. When $N \gg \Delta$[4], we find that

$$\Delta^{(\beta)} \equiv \Delta = N\pi^{\beta+1}\left(\frac{\csc(\frac{\pi}{\beta+1})}{1 + N(\beta+1) + \beta}\right)^{\beta+1}. \tag{29}$$

Plugging this back into our expression for the loss in Equation (21), we find that for any given eigenvalue scaling $\beta$ and $N < D \ll M$,

$$\hat{\mathcal{L}}(N, M, D) \approx \frac{\sigma_w^2}{2}\frac{N}{1 - N/D} \cdot \pi^{\beta+1}\left(\frac{\csc(\frac{\pi}{\beta+1})}{1 + N(\beta+1) + \beta}\right)^{\beta+1}. \tag{30}$$

The comparison of this theoretical prediction of the loss as a function of $D$ to the numerical simulation can be in Figure 19 for different choices of the scaling exponent $\beta$. We see that the predictions get slightly worse for smaller values of $\beta$. This is expected as the numerical simulations must be carried out with some finite but large value of $M$ ($1.2 \times 10^6$ in these plots). As $\beta \to 0$, the approximation in Equation (25) requires a correspondingly larger value of $M$ to correctly capture the tail behavior of Equation (23). We also compare the prediction Equation (30) to numerical data as a function of $N$, for fixed values of $D$ in Figure 20.

This result immediately implies:

---

[3]Note that this approximation requires that $M$ be much larger than any other scale. In particular, when $\beta$ is close to zero, the sum in Equation (24) converges very slowly, and is only approximated by the integral when $M$ is sufficiently large.

[4]The validity of this limit can be argued as follows: note that we can break up the integral in Equation (25) into two regimes: one where the first term in the denominator dominates and one where the second term dominates. The transition point where this happens is at $z = z_0$ where $\Delta z_0^{\beta+1} \approx N$, and so

$$1 = \left|\int_1^{z_0} \frac{dz}{N} + \int_{z_0}^\infty \frac{dz}{\Delta z^{\beta+1}}\right| \leq \left|\int_1^{z_0} \frac{dz}{N}\right| + \left|\int_{z_0}^\infty \frac{dz}{\Delta z^{\beta+1}}\right|. \tag{26}$$

Evaluating, we thus have

$$1 \lesssim \frac{1+\beta}{\beta}\frac{1}{N}\left(\frac{N}{\Delta}\right)^{\frac{1}{\beta+1}} \tag{27}$$

and so

$$\Delta \lesssim CN^{-\beta}, \tag{28}$$

where $C = [(1+\beta)/\beta]^{\beta+1}$.

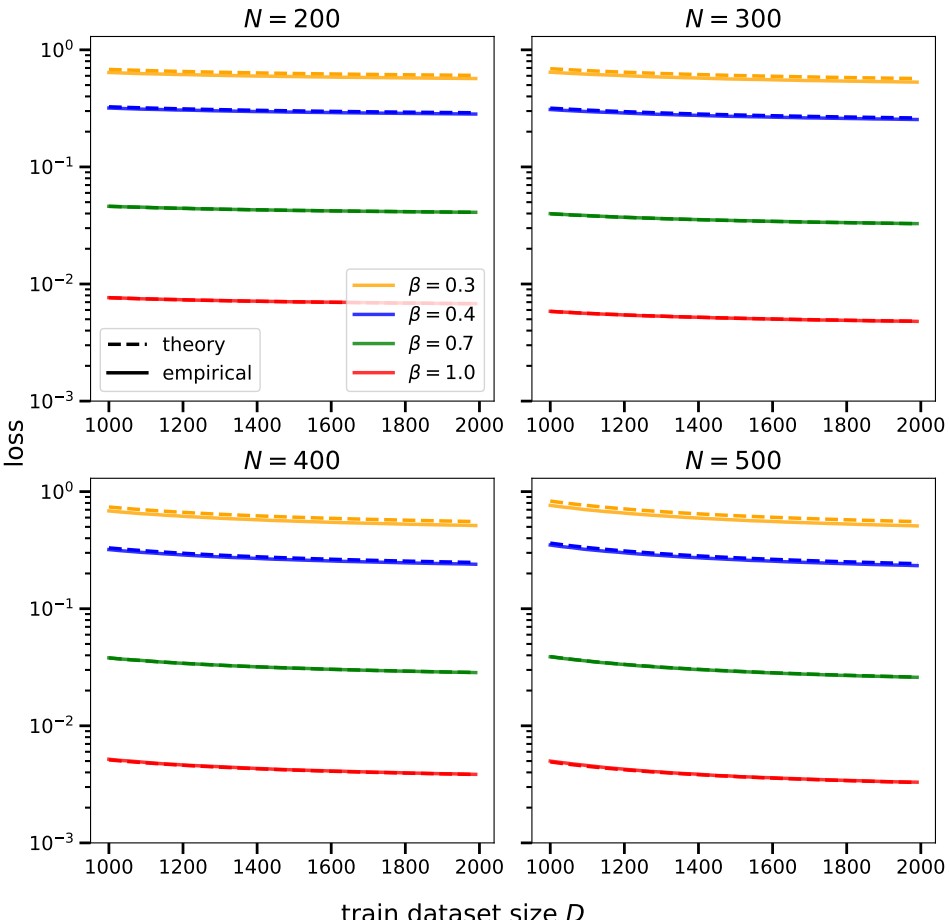

Figure 19: Shows the validation loss plotted as a function of train dataset size for different choices of the eigenvalue scaling $\beta$. Each subplot is a different choice of $N$, the number of model parameters. Solid line indicates numerical data while dashed line indicates theoretical prediction Equation (30). The numerics were carried out with $M = 1.2 \times 10^6$, $\sigma_v = \sigma_w = 1$ and averaged over 2000 random seeds.

- The losses between any two distributions parametrized by eigenvalue scalings $1/i^{\beta+1}$ and $1/i^{\beta'+1}$ for the same values of $N$, $D$, and $M$ will be related to each other via $\mathcal{L}/\mathcal{L}' = \Delta^{(\beta)}/\Delta^{(\beta')}$. Note that this ratio is independent of $D$. We must therefore have that the log-losses on these two distributions will have slope 1 when plotted against each other and intercept $\log \Delta^{(\beta)}/\Delta^{(\beta')}$. This is somewhat different from what we observe in the real datasets, where the slope can be data-dependent (see, e.g., the variation in $\kappa$ across datasets in Table 9.) Nevertheless, the linear model does show that the eigenvalue scaling constrains the behavior of the in-domain losses.

- The dependence of the loss on $N$ and $D$ is not trivial, and does not optically resemble Equation (1) or Equation (3). However, we can study it in different limits to connect it to the usual formulation of scaling laws. In particular, we can expand in the joint limit of $N, D \to \infty$ with the ratio $N/D \ll 1$ fixed. In this limit we find

$$\hat{\mathcal{L}}(N, M, D) \approx \frac{\sigma_w^2}{2} \left( \frac{1}{N^\beta} + \frac{1}{DN^{\beta-1}} + \mathcal{O}(N/D) \right) \frac{1}{(\beta+1)^{\beta+1}} \pi^{\beta+1} \csc(\frac{\pi}{\beta+1})^{\beta+1}. \tag{31}$$

We can see that the term in the parantheses includes a cross term between $N$ and $D$. This cross term is precisely the leading term we would obtain if we expanded a scaling law of the form $\left( \frac{A}{N} + \frac{B}{D} \right)^\beta$ at

large $D$ if $A = 1$ and $B = \beta^{-1}$. This indicates that Equation (2) with $\alpha/\beta = 1$ correctly describes the scaling of this model in the underparametrized regime, consistent with the result presented in Maloney et al. (2022).

Taken together, these results suggest that this theoretical model captures some of the observed phenomena, but that some richer component of the real dataset setting is still missing. In particular, we cannot establish a similar result on train-to-test transfer, since the model manifestly does not capture any information out-of-distribution.

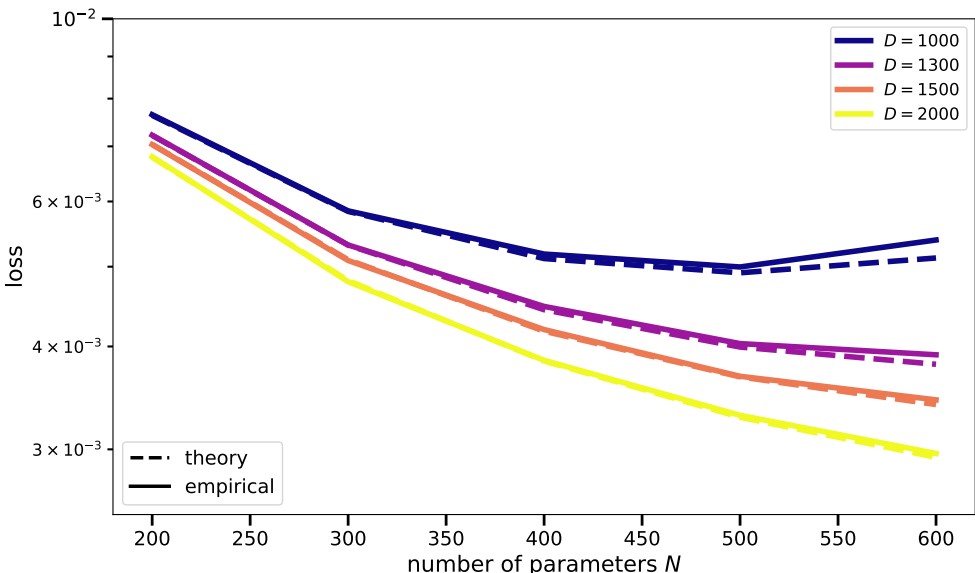

Figure 20: Shows the validation loss as a function of the number of model parameters $N$, for fixed values of the train dataset size $D$ and $\beta = 1$. Solid line indicates numerical data while dashed line indicates theoretical prediction Equation (30), with the same choice of hyperparameters as in Figure 19.

