# OpenReview forum: "Loss-to-Loss Prediction: Scaling Laws for All Datasets"
_TMLR — Accepted by TMLR_

### Review · Reviewer_xxrN · 2024-12-05

**Summary Of Contributions:**

This paper focuses on scaling laws. Instead of focusing on analyzing the scaling properties on a single training data distribution, the authors are analyzing what are the scaling-laws relationships between different training data distribution. They first present results on what they called loss-to-loss relations in which, the power-law relationship is analyzed between different training data distribution for a given compute budget. Then, they analyze the relationship train-to-test when the training dataset is different from the test dataset which is helpful to predict performance on downstream tasks. Lastly, they present test-to-test scaling law relationships where they compare the loss on the same test set while the model are trained on different datasets. The authors ran experiment across several NLP and code datasets. They are able to show that some scaling law properties translate from text data that does not contain code to code data which mean that there approach could generalize even in slight shift in the data distribution.

**Audience:**

Yes

**Claims And Evidence:**

Yes

**Requested Changes:**

No changes requested on my side.

**Strengths And Weaknesses:**

Strengths:
- The paper is very well written and well motivated
- The experimental setup is convincing (variety of datasets and downstream tasks).
- All the hyper-parameters are listed.
- Really appreciate the discussion section. Authors are really straightforward with the limitations of their methods.
- The additional loss to accuracy analysis in the appendix is very valuable.
- Good literature review.

I think this is a solid paper, with evidences supporting the claims and that will be definitively be of interest to the TMLR audience.

---

> ### Author Response · Authors · 2024-12-18
>
> Thanks for the positive assessment! We have made a few updates to the paper in response to the other reviewers, please let us know if there is anything you would like us to change.

---

### Review · Reviewer_4pEG · 2024-12-12

**Summary Of Contributions:**

*Idea*:  The paper presents a method to predict one loss from another (train to test/ train to train/ train to test) and applies this method to predict losses across different pre-training datasets and between pre-training data and a given downstream test dataset.

*Method*: The authors propose a formulation to translate scaling laws between datasets and connect it to a more generic scaling law formulation akin (but more general) to scaling laws in the literature. The authors detail the formulation for three cases: train to train losses, train to test losses and test to test losses, and show that there are simple shifted power law relationships between the different losses in all those cases. Finally, the authors discuss the implications of their approach to better understand scaling laws, transfer learning, and generalization to downstream tasks.

*Experiments*: The authors perform large-scale experiments with 88 models to train in those three settings on 6 pre-training datasets and 11 downstream tasks to determine the coefficients of their scaling laws and they show that leveraging data from multiple pre-training datasets can yield better predictions about the loss when the model is trained on new datasets compared to fitting independent scaling laws.

**Audience:**

Yes

**Claims And Evidence:**

Yes

**Requested Changes:**

No strong requested changes but a small suggestion presented in the weaknesses part.

**Strengths And Weaknesses:**

**Strenghts**

- The paper is well written, with clear notations, and motivations;
- The proposed approach is well explained with thorough empirical validations;
- The related work is comprehensive and covers the literature well;
- The experiments are large-scale and the ablation studies are very complete;
- I particularly appreciate the explanations of the authors regarding their methodological and implementation choices compared to the literature. In addition, they perform the additional experiments needed to justify their choices;
- Finally, the detailed discussion on the limitations and future work helps the reader to properly assess the contributions of the paper and what it brings to the current literature. This is very much appreciated.

**Weaknesses**

Overall, I find the paper very good. I only have one small issue with the current structure that, in my opinion, can "lose" the reader a little bit as the benefits of the approach are not highlighted enough. I think that Sections 4 and 5 lack some transitions to better explain to the reader the benefits of the approach presented in Section 4. I would suggest presenting the generic loss-to-loss prediction of Eq. (9) at the beginning of Section 4 with some explanations of why it matters and then detailing the three cases before starting Section 5 with the practical use cases of the approach. I believe this would strengthen the usefulness of the approach for the reader. What do the authors think about this suggestion? I would appreciate it if the authors could clarify that with me in case I am missing something that would better explain the current organization.

Overall, I think this paper has strong contributions and is very well written which greatly helps in understanding it. It tackles an important problem and tries to provide leverageable takeaways while also discussing its potential shortcomings, which is not done enough in my opinion in the ML field.

---

> ### Author Response · Authors · 2024-12-18
>
> We would like to thank the reviewer for the positive assessment of the paper and constructive comments and questions.
>
> We have restructured the paper to present the general hypothesis first as Section 4 and then go through the evidence in the particular cases in Section 5 before looking at the practical use cases in Section 6. Hopefully this is more clear, and let us know if this seems better to you.

---

> > ### Comment · Reviewer_4pEG · 2024-12-19
> > **Thank You!**
> >
> > I thank the author for their prompt answer and for updating the writing structure. It does seem better to me and I appreciate the effort from the author. The revision has improved the paper which was already very good in my opinion. I maintain my accept the recommendation.

---

### Review · Reviewer_D86T · 2024-12-13

**Summary Of Contributions:**

- This paper explores how scaling laws can be extended to account for changes in data distribution. This approach examines relationships between losses across datasets, aiming to predict one loss from another. The authors identify three main relationships: (1) Train-to-Train: Predicts training loss across two datasets paired by the same compute; (2)  Train-to-Test: Relates training loss on one dataset to test loss on another. (3) Test-to-Test: Links test losses across models trained on different datasets. These relationships are modeled using polynomial functions.
- This paper provides extensive empirical results across diverse datasets, including those with drastically different content (e.g., code vs. non-code datasets) and eleven downstream tasks.
- This paper provides applications for fitting such loss-to-loss curves: (1) Translating scaling laws can optimize predictions for new datasets, reducing the computations. (2) Loss-to-loss prediction could guide data mixing and designing efficient training strategies.

**Audience:**

Yes

**Claims And Evidence:**

Yes

**Requested Changes:**

- It would be better to provide more abundant experimental setup details. Please see the weaknesses above.
- For each curve, it would be better to provide how accurately the curve fits the underlying empirical data points.
- In Section 4.2, are the models fine-tuned on the downstream datasets?
- It would be better to provide algorithmic boxes for methods fitting each curve (in each section). It would help readers to understand the procedure of the paper.
- In Section 5, it would be better to provide the actual computational cost of fitting each curve, for example, GPU hours and FLOPs.

**Strengths And Weaknesses:**

### Strengths

- The paper extends scaling law analysis by studying loss-to-loss prediction. The authors identify consistent loss-to-loss curves across a wide range of pre-training datasets and downstream tasks. This includes datasets with significantly different content (e.g., code vs. non-code).

- By identifying and validating shifted power law relationships, the paper provides a model for predicting losses across distributions, which saves the computation for directly fitting the scaling law curves.

### Weaknesses

- It would be better to provide more details in the experimental setup, as this paper primarily focuses on empirical results. Although they may seem standard, they will help readers interpret results. For example, how are the scaling law curves and loss-to-loss curves fitted? How many models are trained? For a certain FLOP budget and model size, how are the experiments implemented? In addition, there are downstream datasets. What is the setup for the downstream evaluation or fine-tuning?

- It would be better to provide more quantitative results in evaluating scaling laws and loss-to-loss curves. For example, for all the curves reported in the paper, it would be better to define an error metric between the curve and data points (how accurate are the curves)? How many points are used to fit the curves (sample complexity)? Why is the chosen loss-to-loss parameterized function better than other options (as mentioned in Section 4.1)?

- It would be better to provide more analysis from the theoretical perspective. As the scaling laws are closely related to generalization theory, why is one function favored over other functions? Can the authors provide more intuition from the existing generalization theory literature?

---

> ### Author Response · Authors · 2024-12-18
>
> We would like to thank the reviewer for the positive assessment of the paper and constructive comments and questions.
>
> Our updated draft hopefully resolves all of the concerns that are raised. Specifically, all the requested changes have been made:
> - Many of the experimental details are in Section 3.2. We have also re-organized the appendix and added more details to appendix C and D. We added a new appendix F that explains how the curve fitting works in more detail. We also have included in the supplement the notebooks we used to do all the fitting. Upon publication we will also release all the training code and the 588 trained models to support reproducibility.
> - We have added an appendix B.1 with R^2 values and extrapolation predictions for a subset of our curves compared to some simple baselines.
> - The models are never fine-tuned. Evaluation is always zero-shot. We clarified this in section 3.2.
> - This information about how our curve fitting works is in the new appendix F.
> - We have added a note about total FLOPs at the top of section 6.1 (this was deducible from summing across the points in the plots, but we agree that it was hard to parse).
>
> There were also some questions raised about how we chose the functional form of the loss-to-loss prediction. In response to Reviewer 4pEG, we added a new section that more explicitly explains our loss-to-loss prediction hypothesis. This section hopefully answers some of these questions. We will also point the reviewer to Appendix G where we go deep into the theory connections in linear models of scaling laws. Note that traditional generalization theory that only gives upper bounds is not as useful for studying scaling laws as compared to these settings where we can have theory about the exact scaling of the loss. Unfortunately, from our derivations it seems that these models are actually not sufficient to capture the loss-to-loss phenomena, so it seems that more complicated non-linear models may be needed, but this is beyond the scope of our more empirical work.

---

> > ### Comment · Reviewer_D86T · 2024-12-18
> > **Official Comment by Reviewer D86T**
> >
> > Thanks to the authors for their responses! The revision has improved the paper.
> > - I found the content in Appendix G interesting. It would be great if this connection to the theory could be highlighted in the main text. For example, some high-level summary could be added to Section 4 to make the motivation stronger.
> > - The empirical results are very extensive. It would be great to have an online implementation to replicate the results/or walk through the implementation, if published.
> > - Recent studies have shown that using a (linear) surrogate model can extrapolate the model losses trained on subsets of data/tasks. This line of work connects to this paper, while this paper tries to extrapolate the model loss based on the number of training samples and model sizes. It would be better to discuss the connection with this line of works, for example:
> >
> > Datamodels: Predicting Predictions from Training Data. ICML 2022.
> >
> > Identification of Negative Transfers in Multitask Learning Using Surrogate Models. TMLR 2023.

---

> > > ### Author Response · Authors · 2024-12-19
> > >
> > > Thanks for the quick response!
> > >
> > > - Thanks for the suggestion, we added a paragraph to section 4 in a new revision.
> > > - Yes, we will release the entire codebase and trained models on publication.
> > > - Thanks for the pointer, we added a brief discussion of this line of related work in the related work section.

---

### Decision · Action_Editor_Kmwc · 2025-02-06

**Recommendation:** Accept as is

**Comment:**

All the reviewers unanimously agreed to accept this work and stated that it is novel and interesting. The authors have done a great job in experiments and writing.

**Audience:**

This paper is definitely interesting for a large group of ML researchers and should have enough audience from the TMLR community.

**Claims And Evidence:**

This paper proposes methods to predict one loss from another and extends scaling laws to account for changes in data distribution. All three reviewers strongly agree that the work is novel and significant. All the claims in the paper are well supported by experiments, and the authors have added all the extra information the reviewers asked for.